



# Structural and functional control of surface-patch to hillslope-scale runoff and sediment connectivity in Mediterranean-dry reclaimed slope systems

Mariano Moreno-de-las-Heras[1,2]*, Luis Merino-Martín[3,4], Patricia M. Saco[5], Tíscar Espigares[6], Francesc Gallart[1], José M. Nicolau[7]

[1] Institute of Environmental Assessment and Water Research (IDAEA, CSIC), 08034 Barcelona, Spain
[2] Desertification Research Centre (CIDE, CSIC-UV-GV), 46113 Moncada (Valencia), Spain
[3] AMAP, INRA, CIRAD, CNRS, IRD, University of Montpellier, Montpellier, France
[4] CNRS, UMR CEFE, Montpellier, France
[5] Civil, Surveying and Environmental Engineering, The University of Newcastle, 2308 Callaghan (NSW), Australia
[6] Department of Life Sciences, Faculty of Sciences, University of Alcalá, 28805 Alcalá de Henares (Madrid), Spain
[7] Technical School and Environmental Sciences Institute, University of Zaragoza, 22071 Huesca, Spain

*Correspondence to*: Mariano Moreno-de-las-Heras (mariano.moreno@idaea.csic.es)

**Abstract.** Runoff and soil erosion in Mediterranean landscapes are affected by multiple factors that interact at a variety of spatial scales with variable degrees of connection. In these systems, connectivity has emerged as a useful concept for exploring the movement of runoff and sediments between landscape locations and across spatial scales. In this study, we examine the structural and functional controls of surface-patch to hillslope-scale runoff and sediment connectivity in three Mediterranean-dry reclaimed mining slope systems that have different long-term development levels of vegetation and rill networks. Structural connectivity, or the extent to which surface patches that facilitate the production of runoff/sediments are physically linked to one another, was assessed using a flowpath analysis of coupled vegetation distribution and surface topography. Functional connectivity, determined as the spatial continuity of surface fluxes across scales, was further explored using the ratio of surface-patch to hillslope-scale observations of runoff and sediment yield for 21 monitored hydrologically active rainfall events. Event-based (functional) runoff connectivity was found to be dynamically controlled by antecedent precipitation conditions and rainfall intensity and, at the same time, was strongly modulated by the structural connectivity of the slopes. In the absence of rill networks, both runoff and sediments for all events were largely redistributed within the analysed systems, resulting in low functional connectivity. Sediment connectivity increased with rainfall intensity, particularly in the presence of rill networks where active incision under high intensity storm conditions led to large non-linear increases in sediment yield from the surface-patch to the hillslope scales. Overall, our results demonstrate the usefulness of applying structural and functional connectivity metrics for practical applications, and for assessing the complex links and controlling factors that regulate the transference of both runoff and sediment yield across different landscape scales.



## 1 Introduction

Surface processes in Mediterranean landscapes are affected by multiple factors (e.g., rainfall characteristics, soil surface
properties, vegetation, micro-topography, landforms) that interact at a variety of spatial scales (from the soil surface patch to
the plot, hillslope and catchment scales) with variable degree of connection, resulting in intricate responses of runoff and
sediment yield (Puigdefábregas et al., 1999; Calvo-Cases et al., 2003; Cammeraat, 2004; Yair and Raz-Yassif, 2004; Boix-
Fayos et al., 2006; Moreno-de-las-Heras et al., 2010; Mayor et al., 2011; Gallart et al., 2013; Marchamalo et al., 2016). In these
complex systems, connectivity has emerged as a useful concept for studying the movement and transference of runoff and
water-borne materials (e.g., sediments, nutrients, seeds) between landscape locations or scales (Bracken and Croke, 2007;
Wainwright et al., 2011; Bracken et al., 2013; Reaney et al., 2014; Keesstra et al., 2018; Saco et al., 2019).

Two conceptual elements of connectivity that facilitate the analysis of the spatial and temporal dynamics of both runoff
and sediments throughout hillslopes and catchments have been proposed: structural and functional connectivity (Turnbull et
al., 2008; Wainwright et al., 2011; Okin et al., 2015). Structural connectivity refers to the spatial arrangement of hydrologically
significant units or elements, and it captures the extent to which these units are physically linked to each other to allow for the
transfer of water and sediment fluxes. The second, functional (or process-based; Bracken et al., 2013) connectivity, refers to
the generation of active connections of runoff and/or sediment pathways during a particular rainfall event. In the case of runoff,
functional connectivity depends on the dynamics of overland flow generation, routing, and downward re-infiltration, while for
sediments functional connectivity is a function of the detachment, entrainment, deposition and remobilization of sediments
across scales (Wainwright et al., 2011; Bracken et al., 2013; Turnbull and Wainwright, 2019).

Multiple studies have focused on the analysis of the effects of landscape structural components of connectivity on runoff
and soil erosion, particularly in Mediterranean-dry and other dryland systems. For example, in water-limited environments
with patchy vegetation, measures related to the spatial organization (i.e., the pattern, patch size and landscape position) of
vegetation explain runoff and soil erosion better than average vegetation cover (Bautista et al., 2007; Arnau-Rosalen et al.,
2008; Puigdefábregas, 2005). In fact, the spatial arrangement of surface features (e.g., vegetation cover, rills, gullies, channels,
terraces) controls the distribution of source and sink elements in these landscapes from a structural perspective, largely driving
the production and transference of runoff and sediments across scales (Cammeraat, 2004; Lesschen et al., 2009; Turnbull et
al., 2010; Merino-Martín et al., 2015; Marchamalo et al., 2016; Moreno-de-las-Heras et al., 2019). Structural connectivity can
be highly dynamic over long time periods (e.g., decades or longer) as a result of changes in vegetation, land use and surface
morphology. However, structural connectivity is generally considered a static landscape feature over the time periods of
interest (e.g., the hydrological year), which has facilitated the application of this concept in hydrological and geomorphological
studies using surface contiguity indexes (Heckmann et al., 2018 and references therein).

The interactions between the structural connectivity of a landscape and the strength of the transport vector (e.g., a flood)
determines functional connectivity (Wainwright et al., 2011; Bracken et al., 2013; Reaney et al., 2014; Okin et al., 2015).
Surface patches respond to rainfall characteristics and (antecedent) soil moisture conditions to determine the production and



transport of both runoff and sediments through the landscape (Cantón et al., 2011; Mayor et al., 2011; Rodríguez-Caballero et al., 2014). Thus, functional connectivity may dynamically vary between rainfall events. There is little consensus, however, on how to quantify functional connectivity (Okin et al., 2015). Several research approaches have been applied to analyse functional aspects of connectivity in terms of the continuity of runoff and sediment fluxes along hillslopes, landscapes and

catchments. These approaches cover a wide array of methods including hierarchical (i.e., nested, stratified and/or scaled) runoff/sediment yield measurements (Cammeraat, 2004; Yair and Raz-Yassif, 2004; Moreno-de-las-Heras et al., 2010; Mayor et al., 2011), field-based mapping and modelling of active runoff/sediment flowpaths (Arnau-Rosalen et al., 2008; Marchamalo et al., 2016; Turnbull and Wainwrigth, 2019), fallout radionuclide and rare-earth element sediment tracing applications (Masselink et al., 2017a; Moreno-de-las-Heras et al., 2018) or particle-in-motion tracers and overland flow sensors (Hardy et

al., 2017; Masselink et al., 2017b).

Mediterranean-dry reclaimed mining slope systems are characterized by the local convergence of high storm erosivity, poorly developed soils, scarce vegetation cover, and rough topography. These characteristics can lead to the genesis of important amounts of overland flow, promoting soil erosion processes, which typically lead to rill and gully development (Nicolau and Asensio, 2000; Nicolau, 2002; Moreno-de-las-Heras et al., 2009; Martín-Moreno et al., 2018). The analysis of

runoff and sediment connectivity has a critical relevance for landscape management in these human-made, water-limited environments, where the functional components of runoff and sediment connectivity (e.g., the routing of runoff and sediment fluxes from the surface patch to the broader, hillslope scale) can shape on-site structural connectivity factors (e.g., vegetation patterns, spatial distribution of rill networks and sedimentation areas) over long periods, conditioning the long-term eco-geomorphic stability of the reclaimed systems (Moreno-de-las-Heras et al., 2011a). In fact, within-slope spatial redistribution

of runoff and sediment fluxes in these reclaimed Mediterranean-dry systems feeds back into patch-scale hydrological behaviour by controlling the availability of water and soil resources for the long-term development of vegetation cover (Espigares et al., 2011; Moreno-de-las-Heras et al., 2011b; Merino-Martín et al., 2015). Furthermore, the magnitude and cross-scale transmission of runoff and sediments in these reclaimed Mediterranean systems largely determines their off-site effects in the form of runoff and sediment conveyance to downstream channels and environments (Martín-Moreno et al., 2018).

In this study, we apply the concepts of structural and functional connectivity to analyse the factors that control the transference of runoff and sediments from the surface patch to the hillslope scale in three reclaimed Mediterranean-dry systems that differ in their vegetation organization and landform features (i.e., rill networks). Our assessment is based on the analysis of patch-to-hillslope runoff/sediment flow continuity of 21 active events monitored during 2007-08 using a hierarchical (scaled) measurement approach (Merino-Martín et al., 2012a). Specifically, our analysis aims to determine how structural

components of the three reclaimed slope systems (i.e., the spatial distribution of vegetation cover and micro-topography, including rill networks) dynamically interact with rainfall characteristics (i.e., storm depth, rainfall duration and intensity) and antecedent storm conditions to generate spatially continuous runoff and sediment fluxes.



## 2. Materials and Methods

### 2.1 Study Area

This work was carried out in the Utrillas field site (Figure 1a), an experimental station located in "El Moral", a reclaimed surface coalmine in central-eastern Spain (40º47'24'' N, 0º49'48'' W, 1100 m). The climate is Mediterranean-dry, with a mean annual air temperature of 11ºC. Mean annual precipitation is 450 mm, most of which occurs in spring and autumn. Hargreaves and Samani (1985) potential evapotranspiration is around 900 mm, yielding a hydrological deficit of approx. 450 mm that is concentrated in the summer (López-Martín et al., 2007). The average number of precipitation events in the area is 50-70 per

year. Especially remarkable is the rainfall erosivity of high-intensity late-spring and summer convective thunderstorms (<10% rainfall events), which can reach up to 100 mm rainfall in 24h (Peña et al., 2002).

The study site encompasses three experimental slopes, all north facing with a general gradient of about 20º (Figures 1b and 1d). The slopes were reclaimed during 1987-89 with the following procedure. First, a 100-cm-thick layer of clay-loam overburden substratum was spread over the slopes. Next, the surface was prepared for re-vegetation by applying cross-slope

ploughing to create a transversal pattern of surface roughness that would facilitate water storage and infiltration. Finally, the slopes were sown with a seed mixture of perennial grasses and leguminous herbs (*Festuca rubra*, *Festuca arundinacea*, *Poa pratensis*, *Lolium perenne*, *Medicago sativa* and *Onobrychis viciifolia*).

Although the three slopes were originally restored using the same procedure, their subsequent evolution and ecosystem recovery level displayed differences due to variations in their geomorphological design, particularly in the upper section of the

slopes (Moreno-de-las-Heras et al., 2011a; Merino-Martin et al., 2015). Specifically, these variations occurred due to the existence of a bare soil, 40º-steep berm integrated at the top of two of the experimental slopes (Slopes 1 and 2, with berm sizes of 50 and 20 m$^2$, respectively; Figure 1b) that generates important amounts of overland flow, early promoting intense soil erosion (Merino-Martin et al., 2012a). These mechanisms resulted in the formation of a deeply incised (up to 35 cm depth in the middle and lower sections of the slope) and fairly dense (0.6 m per m$^2$ density) rill network in Slope 1 –where the overland

flow contributing berm reaches its largest size– and, also, important variations in vegetation development among the three experimental slopes (Figure 1d). At the time of data collection for the present study (from October 2007 to December 2008, after 20 years of dynamic evolution from initial reclamation) the three experimental slopes showed different levels of vegetation development (30%, 45% and 55% cover for Slopes 1, 2 and 3, respectively), soil erosion intensity (2007-08 sediment yield was 1824, 81 and 4 g m$^{-2}$ for Slopes 1, 2 and 3, respectively) and runoff production (2007-08 runoff coefficient

was 14.5%, 2.1% and 0.4% for Slopes 1, 2 and 3, respectively; Table S1 in the online supplement).

### 2.2 Experimental layout: 2007-08 hydrological measurements

The analysis of the connectivity of water and sediment fluxes can be assisted by the use of hierarchical approaches at a range of scales, which are useful for capturing the spatial and temporal patterns of both runoff and sediment generation and redistribution (Boix-Fayos et al., 2006; Sidle et al., 2017). We monitored runoff and soil erosion in the experimental slopes





from October 2007 to December 2008 by applying a scaled approach (Merino-Martín et al., 2012a). This approach included naturally delimited runoff/erosion plots distributed at two different scales (Figure 1c): (a) the hillslope scale and (b) the surface-patch scale.

Three broad, hillslope-scale runoff/erosion plots were installed in naturally delimited catchments in the experimental slopes (Figure 1d: catchment areas are 498, 511 and 1474 m² for slopes 1, 2 and 3, respectively). At the foot of each catchment, two

collectors and a central cemented outlet were installed. From the outlet, runoff was routed through a pipe into 200-litre storage tanks connected by multi-slot runoff Geib (1933)-type dividers (Figure 1d).

At the patch scale, seven different soil surface types were identified within the experimental slopes as a function of vegetation community composition and soil surface traits (e.g., dominant plant species, species richness, cover, presence of soil surface crusts and small pedestals). The soil surface patch types (Figure 1f; Table S1 in the online supplement) included:

barely covered areas (<5% cover) with scattered clumps of perennial forbs dominated by *Medicago sativa* (Ms); sparsely covered areas (cover approx. 30%) with grasses dominated by *Dactylis glomerata* (Dg), or with dwarf shrubs dominated by either *Santolina chamaecyparissus* (Sch) or *Thymus vulgaris* (Tv); and finally, densely covered areas (>70% cover) dominated by grass species (*Lolium perenne*, Lp, and *Brachypodium retusum*, Br) or by shrubs (*Genista Scorpius*, Gs). In order to monitor runoff and sediment yield at the patch scale, 27 Gerlach (1967) troughs (each 0.5 m wide and connected to 100-litre drums for

runoff and sediment storage; Figure 1e) were distributed in the slopes between the seven surface types. The spatial organization of the surface types and contributing area of the Gerlach troughs in the experimental slopes was determined in the field using a total station (Topcon GTS212). Catchment area of the Gerlach troughs, delimitated by surface micro-topography and vegetation barriers, ranged from 1 to 16 m² (Table S1 in the online supplement).

Runoff amount was measured in the storage tanks/drums within a day after each runoff event (runoff-producing events

occurring within a 24 h period were considered to belong to the same event). The stored runoff was stirred, and 1-litre representative samples were taken. Sediment concentrations were determined by oven-drying the collected runoff samples (at 105°C) until a constant weight was achieved.

Rainfall depth (Dp, mm) for each event was measured using three bulk precipitation collectors located in the experimental slopes. Rainfall duration (Rd, h) and both 15- and 30-min maximum rainfall intensities ($I_{15}$ and $I_{30}$, respectively, mm h$^{-1}$) were

measured using an automated recording raingauge (Davis GroWeather) installed in the experimental station. Mean rainfall intensity (Im, mm h$^{-1}$) for each event was calculated as the ratio of total precipitation to rainfall duration. In order to characterize the antecedent rainfall conditions of the events, we used the Antecedent Precipitation Index (API, mm; Kohler and Linsley, 1951). API is calculated as:

$$API = \sum_{t=-1}^{-T} P_t k^{-t} \tag{1}$$

where, $P_t$ (mm) is the precipitation on a given day $t$; $k$ is a dimensionless decay coefficient that represents a measure of the declining influence of past precipitation on current soil moisture state; and $T$ (days) is the antecedent period considered for the calculation of the index. We used fortnightly soil moisture records obtained from autumn 2007 to winter 2008 in the experimental slopes (3x12 TDR profiles of 50-cm depth; Merino-Martín et al. 2015) to parameterize API for this study ($k=$



0.98, $T$= 10 days; complimentary field-calibration details of API parameters in Supplementary Methods S1 of the online
supplement).

A total of 21 rainfall events out of 74 precipitations produced runoff/erosion at the patch scale during the study period
(October 2007-December 2008, total rainfall 703 mm), although only 17 generated significant hydrological responses at the
hillslope scale (Merino-Martin et al. 2012a). We used the complete set of 21 hydrologically active events for this study (full
hydrological data is available in Table S2 of the online supplement).

## 2.3 Structural connectivity quantification: distribution of sources and sinks

Previous research carried out in the Utrillas field site applying small-scale (0.25 m$^2$) rainfall simulations indicated that surface
patches with vegetation cover under 30-50% can generate important amounts of runoff/sediments, thus acting as "sources" of
water runoff and sediments within the slopes. Conversely, surface patches with vegetation cover above 50% regulate soil
surface hydrological responses very efficiently and may also operate as flow obstructions, behaving as "sinks" of runoff and
sediments (Moreno-de-las-Heras et al., 2009). Structural connectivity for this study (i.e., the physical linkage of
runoff/sediment source areas within the experimental slopes) has been quantified from coupled analysis of binary maps of
vegetation density (above/below 50% cover) derived from high-resolution multispectral aerial photography and field-based
digital elevation models (DEMs) of the experimental slopes.

Multispectral information of the experimental slopes was obtained from a high-resolution four-band aerial image (0.5-m
pixel resolution) captured by the Spanish National Plan for Aerial Orthophotography (PNOA, Spanish National Geographic
Institute) in late spring 2009. We used the red and near-infrared bands to generate raster maps of the Normalized Difference
Vegetation Index (NDVI). NDVI is a chlorophyll-sensitive vegetation index that strongly correlates with vegetation cover and
green biomass density (Anderson et al., 1993). We applied field based NDVI thresholding (Scanlon et al., 2007) to transform
the raster NDVI maps of the experimental slopes into binary maps of sinks and sources of runoff and sediments. To this end,
we first used reference vegetation density data collected in the field (3x35 quadrats of 0.25-m$^2$ size regularly distributed within
the experimental slopes; Merino-Martin et al. 2012b) for determining the proportional abundance of sink areas with above
50% vegetation cover for each analysed slope system. We then classified the pixels in each slope by thresholding the NDVI
values in the raster maps to match the ground-based proportional abundance of sink areas obtained in the first step. The
application of this image processing methodology resulted in the generation of a high resolution (0.5 m per pixel) binary map
product representing the distribution of sinks and sources (>50% and <50% cover patches, respectively) of runoff and
sediments for each experimental slope.

Detailed digital elevation data for the analysis of structural connectivity was obtained from a topographical field survey
(Merino-Martín et al., 2015). DEM break lines and filling points (~0.5 points m$^{-2}$ data density) were obtained using a total
station (Topcon GTS212). Scattered elevation data was interpolated using thin plate splines to match the (0.5-m resolution)
grid-based binary maps of sources and sinks of the experimental slopes.



We used the flowlength calculator developed by Mayor et al. (2008) along with the source/sink binary maps and the obtained DEM data to analyse the physical linkage of runoff/sediment source areas in the experimental slopes. This calculator applies a D8 flow routing algorithm (O'Callaghan and Marks, 1984) to determine the length of the runoff paths in the downslope direction until a sink is reached in the binary map (i.e., >50% vegetation cover pixel) or the outlet of the system is

reached. Flowpath measurements were standardized to obtain a structural connectivity indicator (Sc; dimensionless). Standardization was carried out for every pixel in the raster maps of the experimental slopes by determining the ratio of the calculated flowlength to the (topography-based) downslope distance to the outlet of the system. Sc values range from 0 (sink pixels) to 1 (source pixels connected to the outlet of the slope system without the interference of any sinks).

For each experimental slope, we calculated the cumulative probability distribution function (CDF) of the structural

connectivity (Sc) values. Differences between the three experimental slopes on the Sc probability distribution were analysed using two-sample Kolmogorov-Smirnov tests. Mean Sc values (hereafter S̄c) were calculated for each slope system as an integrative indicator of the structural connectivity at the hillslope-scale level.

**2.4 Functional connectivity quantification: flow continuity across scales**

The study of variations of surface hydrological responses (i.e., runoff and sediment yield) at a range of scales provides a tool

for the assessment of the dominant processes regulating the routing of runoff and sediments (e.g., flow concentration, transmission losses among landscape elements, downslope runoff re-infiltration, sedimentation) across the landscape (Cammeraat, 2004; Yair and Raz-Yassif, 2004; Moreno-de-las-Heras et al., 2010; Mayor et al. 2011; Sidle et al., 2017). To determine the spatial continuity (or functional connectivity) of both runoff and sediment fluxes across scales (i.e., from the surface-patch to the hillslope scale) in the experimental slopes, we applied a two-step approach. First, for each event and

experimental slope we determined the integrated patch-scale response of runoff and sediment yield. This integrated patch-scale response was computed by weighing the surface-patch runoff and sediment observations (from the data recorded in the Gerlach troughs) with the proportional distribution (i.e., percentage area) of the different soil surface types monitored in the slope systems. Second, functional connectivity, assessed as the cross-scale flow continuity of runoff and sediment fluxes ($C_R$ and $C_S$, respectively), was quantified for each event and experimental slope as the ratio of the hydrological and sediment

observations recorded in the broad, hillslope-scale plots to the determined, integrated patch-scale responses of runoff and sediment yield.

The functional connectivity of runoff ($C_R$, dimensionless) ranges between values 0 and 1. $C_R$ equals 0 when there is complete within-slope spatial redistribution of runoff (i.e., no runoff generated at the surface-patch scale reaches the outlet of the slope system). $C_R$ increases as runoff redistribution decreases across scales, and equals 1 when all runoff generated at the

patch scale reaches the outlet of the slope system. Previous research in our study site suggests that any further contributions to hillslope runoff production (e.g., subsurface return fluxes captured by the rill networks) have a marginal impact on the hydrological response of these water-limited systems (Nicolau, 2002; Moreno-de-las-Heras et al., 2010; Merino-Martín et al., 2012a).



The functional connectivity of sediments ($C_S$, dimensionless) decreases from 1 to 0 when significant amounts of sediments

generated at the patch scale are deposited within the slope system before reaching the outlet. However, $C_S$ may take values

over 1 if active rill incision takes place in the experimental slopes, causing the entrainment of significant amounts of sediments

between the patch and hillslope scales (Moreno-de-las-Heras et al., 2010).

Differences between the three experimental slopes on the connectivity of runoff ($C_R$) and sediments ($C_S$) were tested for

the set of 21 hydrologically active events recorded during the study period (October 2007-December 2008) using Kruskal-

Wallis ANOVA. In addition, the broad, hillslope-scale hydrological (sediment) responses of the experimental slopes were

tested against the analysed, per-event functional connectivity of the studied systems by determining the best fitting regression

function linking the determined runoff (sediment) connectivity values and the observed hillslope runoff coefficients (soil

losses).

## 2.5 Functional connectivity quantification: flow continuity across scales

We applied the general descriptors of precipitation characteristics (Dp, Rd, $I_{15}$, $I_{30}$ and Im), antecedent rainfall conditions (i.e.,

the field-calibrated API values), and the obtained $\bar{S}c$ index of structural connectivity to determine the main controlling factors

that drive the functional responses of runoff and sediment connectivity in the experimental slopes for the studied (21

hydrologically-active) runoff/erosion events. We applied a step-by-step model building procedure, using general linear models

(GLM; Christensen, 2002) with $C_R$ and $C_S$ as the outcome, dependent variables. First, an exploratory pre-screening analysis

of the dynamic relationships between the functional connectivity indexes ($C_R$ and $C_S$) and both the rainfall characteristics and

antecedent precipitation conditions was performed using Spearman's R correlations. Second, the pre-screened variables that

showed significant correlations (at $\alpha=0.05$) with the $C_R$ and $C_S$ values of the storms were further applied to model the surface-

patch to hillslope-scale transfer (or flow continuity) of runoff and sediments.

In order to identify the set of explanatory variables that produce the best model for $C_R$ and $C_S$ prediction, alternative GLM

configurations were compared, using a backward model selection procedure. These alternative GLM configurations included

(i) $\bar{S}c$ as a factor with three levels representing the structural connectivity of the three experimental slopes, (ii) all the possible

combinations of significant (Spearman's R) pre-screened variables of rainfall characteristics and antecedent conditions, and

(iii) the interaction terms between $\bar{S}c$ and the pre-screened variables included in each comparison of model structure. The

Akaike Information Criterion (AIC; Akaike, 1974) and the adjusted coefficient of determination (Adj $R^2$), which represent a

trade-off between model complexity and goodness of fit, were used to select the best model for $C_R$ and $C_S$ prediction. Finally,

the model root-mean-squared error (RMSE), and both the effect size (eta-squared values, $\eta^2$) and significance of the model

predictors were evaluated for the selected, optimal GLM configurations. While $C_S$ may take values (largely) above 1 when

active rilling takes place, $C_R$ is constrained to values ≤1 and consequently, may asymptotically approach 1 as rainfall increases.

This fact violates the GLM assumption of linearity for large values of the model predictors. We, therefore, applied logarithmic

transformation to the (Dp, Rd, $I_{15}$, $I_{30}$, Im and API) co-variables to comply with the GLM assumptions for $C_R$ modelling.



## 3. Results

### 3.1 Source/sink distribution and structural connectivity of the slopes

Runoff/sediment sinks (areas with vegetation cover >50%) in Slope 1 were particularly concentrated in its central section, mainly distributed as grass patches dominated by *Lolium perenne* (Lp soil surface patch type; Figure 2a). A well-developed
rill network (density 0.6 m m$^{-2}$) linked the runoff/sediment source areas (<50% veg. cover) located at the top of the experimental slope with both the source areas distributed at the bottom of the slope and the outlet of the system. Similarly, runoff/sediment sink areas for Slope 2 (Figure 2b) were preferentially distributed in the central part of the slope system, mostly in the form of densely vegetated grass and shrub patches (surface patch types Lp, Br, Dg and Gs). However, runoff/sediment source areas at the top of Slope 2 were not physically linked with the source areas distributed at the bottom of this experimental
slope. Finally, sink areas for Slope 3 were broadly distributed in the form of densely vegetated shrub clumps (Gs and Sch surface patch types) within the central and lower sections of the slope system (Figure 2c), largely interfering the connection of the source areas distributed at the top of the slope with the outlet of the system.

Figure 2d shows the cumulative probability distribution function (CDF) of the structural connectivity metric (Sc) in the three experimental slopes, along with their mean values (S̄c). Two-sample Kolmogorov-Smirnov tests indicated that the Sc
CDFs significantly differed between the three experimental slopes at α=0.01. For the rilled system (Slope 1), the probability of finding runoff/sediment source areas physically linked to the outlet of the slope system (Probability Sc ≥ 1) was almost 40%, leading to a large structural connectivity at the hillslope level (S̄c=0.47). Conversely, the abundance of barely covered, source areas connected with the outlets of the two non-rilled slope systems was substantially lower (12% and <1% for Slopes 2 and 3, respectively). This effect was particularly noticeable for Slope 3, where the lack of physical contiguity between the
major runoff/sediment source areas and the bottom of the slope system results in a very low structural connectivity at the hillslope level (S̄c=0.02).

### 3.2 Functional connectivity: cross-scale continuity of runoff and sediments

Functional connectivity or flow continuity of runoff across scales, assessed as the ratio of hillslope-scale to the (proportionally weighed) integrated patch-scale runoff observations, showed important differences for the three experimental slopes.
Cumulative runoff production along the study period (2007-08) decreased from the surface-patch to the hillslope-scale in the three systems (Figure 3a). However, variations in runoff production across scales showed remarkable differences between the slopes. For the rilled slope system (Slope 1), the cross-scale connectivity of 2007-08 cumulative runoff was 0.72, indicating that about 70% of the runoff that was generated at the patch-scale level during the study period effectively reached the outlet of the system (in other words, 30% runoff was redistributed or re-infiltrated within the slope during 2007-08). Differently, less
than 20% of the patch-scale runoff reached the outlets of the non-rilled systems (i.e., connectivity of cumulative runoff was < 0.2 for Slope 2 and 3; Figure 3a).





Runoff connectivity displayed an important variability among the various monitored events ($C_R$; Figure 3c). The values of runoff connectivity across scales in Slope 1 ranged from 0 to around 1 for the 21 active events. In particular rainfall events there was complete runoff redistribution within this slope (i.e., $C_R$=0), while in other events all runoff generated at the patch-

scale level reached the outlet of the system (i.e., $C_R$ about 1). In the case of Slopes 2 and 3, the maximum and mean values of runoff connectivity were notably smaller than for Slope 1, indicating that for all the events, larger fractions of patch-scale runoff were spatially redistributed (i.e., downslope re-infiltrated) within the two non-rilled systems.

Functional connectivity of cumulative (2007-08) sediment yields also showed important differences among the experimental slopes (Figure 3b). For the rilled system (Slope 1), cumulative sediment yield at the hillslope scale was 1.6 times

bigger than patch-scale sediment production. Conversely, the continuity of sediment fluxes across scales was very low for the non-rilled slopes: less than 15% cumulative sediments generated at the patch-scale level in Slopes 2 and 3 during the study period reached the outlet of these non-rilled slope systems.

At the rainfall-event level, the cross-scale continuity of sediment fluxes showed a large variability ($C_S$; Figure 3d). This effect was especially important for the rilled system (Slope 1), where there was complete spatial redistribution of sediments

for particular events (i.e., $C_S$=0, indicating that no patch-scale generated sediments reached the outlet of the system) while for other events –with active rill incision– hillslope-scale sediment yield was up to 5 times bigger than patch-scale sediment production. Flow continuity of sediments across scales for the non-rilled slope systems (Slopes 2 and 3) was low for all the recorded events (maximum sediment connectivity was 0.4 and 0.07 for Slopes 2 and 3, respectively), showing substantial sediment deposition from the patch to the hillslope scales.

Per-event hillslope runoff coefficient and soil erosion increased non-linearly with increasing runoff ($C_R$) and sediment ($C_S$) connectivity, respectively. Hillslope runoff production showed slight variations with cross-scale runoff connectivity up to $C_R$ values about 0.5 (50% runoff redistribution between the patch and hillslope scales) over which, the runoff coefficients of the experimental slopes increased very rapidly (from ~5% to about 30%; Figure 3e). Similarly, hillslope soil erosion showed little change with $C_S$ <1, when active within-slope sediment deposition took place (Figure 3d). However, for $C_S$ values above 1,

when the rill networks actively contributed with fresh sediments to the flow, broad hillslope-scale soil erosion increased very strongly, up to two orders of magnitude. These very high $C_S$ values, observed during extreme sediment response events ($C_S$ peaking up to nearly 5; Figure 3d), suggest that erosion from the rill networks can contribute with up to 4 times more sediments to the outlet than the poorly covered (source) surface patches acting in the experimental slopes.

### 3.3 Impact of structural and dynamic factors on runoff and sediment connectivity

Exploratory analysis of the relationship between runoff connectivity ($C_R$) and the event characteristics indicated that antecedent precipitation (API), both maximum ($I_{15}$, $I_{30}$) and mean ($I_m$) rainfall intensity, as well as storm depth (Dp) significantly correlated with the continuity of runoff across scales (Table 1). These correlations were particularly strong for antecedent precipitation in all the analysed slope systems (0.75-0.80 Spearman's R). GLM modelling of observed $C_R$ values using the pool of significant event characteristics (API, $I_{15}$, $I_{30}$, $I_m$ and Dp) pointed to the effects of (log-transformed) API and $I_{30}$





variables, hillslope structural connectivity ($\bar{S}c$) and their corresponding interaction terms ($\bar{S}c$:API and $\bar{S}c$:$I_{30}$) as the best model structure predictors ($R^2$=0.81, Adj $R^2$=0.78, NRMSE=12%; Figure 4a). No additional increments of GLM complexity resulted in significant improvements of explained $C_R$ variance (Adj $R^2$ values of alternative models in Table S3 of the online supplement). The eta-squared values for the optimal $C_R$ model ($\eta^2$; Figure 4a) revealed a primary influence of hillslope structural connectivity, which absorbed 44.1% and 16.8% of $C_R$ variance in the form of direct ($\bar{S}c$) and interaction ($\bar{S}c$:API

and $\bar{S}c$:$I_{30}$) effects, respectively. The direct effects of the event-driven (API and $I_{30}$) variables accounted for an additional 20% of the observed $C_R$ variance (12.9% and 7.1% for API and $I_{30}$, respectively).

    An increase in both antecedent precipitation and 30-min maximum rainfall intensity led to non-linear increases in runoff connectivity for all the analysed slope systems (Figures 4b and c). $C_R$ increased rapidly for values within 0-20 mm h$^{-1}$ maximum rainfall intensity and 0-50 mm antecedent precipitation, tending to saturate for larger $I_{30}$ and API values. However, the

structural connectivity ($\bar{S}c$) of the analysed slope systems exerted a strong control on these effects. Both maximum rainfall intensity and antecedent precipitation increased runoff connectivity very little for Slopes 2 and 3, as a result of the large within-slope redistribution of runoff that prevailed under all rainfall conditions for these poorly connected ($\bar{S}c \leq 0.17$) systems. Instead, $C_R$ strongly increased with $I_{30}$ and API for the rilled slope system (Slope 1), where runoff producing low-cover areas showed a large spatial contiguity ($\bar{S}c$=0.47). Furthermore, antecedent precipitation had a higher impact compared to the influence of

maximum rainfall intensity on runoff connectivity. In fact, $I_{30}$ displayed a limited impact on runoff connectivity under dry antecedent conditions (e.g., if API=5.1 mm, $C_R$ in Slope 1 can increase up to ~0.4 for large $I_{30}$ values; Figure 4b). Nevertheless, antecedent precipitation showed an efficient capacity to increase runoff connectivity, even for moderate and low intensity rainfall events (e.g., if $I_{30}$=3.4 mm h$^{-1}$, $C_R$ in Slope 1 may grow up to ~0.8 for large API values; Figure 4c).

    The continuity of sediment fluxes across scales ($C_S$) strongly correlated with mean ($Im$) and maximum ($I_{15}$, $I_{30}$) rainfall

intensity, particularly for Slopes 1 and 2 (Table 1). Although considerably less intense, we also found significant correlations between $C_S$ and antecedent precipitation (API). GLM modelling of $C_S$ using these rainfall variables ($Im$, $I_{15}$, $I_{30}$ and API) identified mean rainfall intensity ($Im$), hillslope structural connectivity ($\bar{S}c$) and their corresponding interaction term ($\bar{S}c$:$Im$) as best model structure contributors for predicting sediment connectivity ($R^2$=0.81, Adj $R^2$=0.79, NRMSE=8%; Figure 5a). No additional increments of GLM complexity produced significant improvements of explained $C_S$ variance (Adj $R^2$ values of

alternative models in Table S4 of the online supplement). Similarly to the best $C_R$ model, the $\eta^2$ values for the optimal $C_S$ model indicated a key influence of hillslope structural connectivity, which explained 38.0% and 26.8% of $C_S$ variance for its direct ($\bar{S}c$) and interaction ($\bar{S}c$:$Im$) effects, respectively (Figure 5a). The direct effect of mean rainfall intensity explained an additional 16.1% of the observed $C_S$ variance.

    $C_S$ increased linearly with mean rainfall intensity ($Im$) for the three experimental slopes (Figure 5b). However, the impact

of rainfall intensity on the patch- to hillslope-scale continuity of sediment fluxes was highly dependent on the structural connectivity of these slope systems. For Slopes 2 and 3, with poorly connected low cover areas that act as sediment sources ($\bar{S}c \leq 0.17$), increases of rainfall intensity along the observed $Im$ range (0-6 mm h$^{-1}$) resulted in very small increases in sediment connectivity, pointing to a large within-slope redistribution of sediments for all the explored rainfall conditions. Very





differently, for Slope 1, with a well-developed rill network that provides good structural connection of low cover areas
($\bar{S}c=0.47$) and within-slope conditions for channel incision, $C_S$ largely increased over 1 along the range of observed mean
rainfall intensities, therefore reflecting large increases of unit-area sediment yield from the patch to the hillslope scales under
high intensity rainfall.

**4 Discussion**

Connectivity and scaling are key aspects for the understanding of hydrological and geomorphological processes in the
continuum from small plots to hillslopes and catchments (Bracken and Croke et al., 2007; Sidle et al., 2017). In this study we
shift from the very active, present conceptual discussion of the connectivity theory and their derived hydro-geomorphic study
approaches (Wainwright et al., 2011; Bracken et al., 2013; Okin et al., 2015; Heckmann et al. 2018; Keesstra et al., 2018; Saco
et al., 2019) to the practical application of the concepts of structural and functional connectivity for the analysis of surface-
patch to hillslope scale continuity of runoff and sediment fluxes in Mediterranean-dry reclaimed slope systems.

**4.1 Structural connectivity: organization of vegetation patterns and rill networks**

Both vegetation distribution, which largely influences the spatial organization of patch hydro-sedimentary behaviour, and
surface topography, which controls water and sediment flow direction, represent the major determinants for the potential
transfer of water and sediments in the flowpath-based approach of structural connectivity applied in our study. Dryland
vegetation is frequently organized in patches, ranging from barely to densely covered surfaces, which interact as interconnected
source and sink areas of runoff and sediments (Puigdefabregas, 2005; Saco et al., 2019). This source/sink behaviour largely
controls the within-slope retention of water and soil resources, and has been extensively described as a key structural control
for the production and routing of runoff and sediments in both natural Mediterranean semiarid landscapes (Puigdefabregas et
al., 1999; Cammeraat, 2004; Arnau-Rosalen et al., 2008; Mayor et al., 2011) and reclaimed Mediterranean-dry systems
(Moreno-de-las-Heras et al., 2009; Merino-Martín, 2012b, 2015; Espigares et al., 2013). Our results reveal that hillslope
position of densely vegetation patches is a significant factor affecting the structural connectivity of the analysed reclaimed
slope systems. In our study, the preferential concentration of densely vegetated, sink patches in the middle and lower sections
of Slope 3 dramatically reduces the connectivity of source areas as compared to Slope 2, where the bottom of the slope is
dominated mainly by poorly covered areas (Fig. 2b-d). These results agree with other empirical studies in the Mediterranean
region that indicate that the presence of dense vegetation in and near the lower sections of plots and hillslopes provides a strong
structural control for runoff and sediment delivery (Bautista et al., 2007; Boix-Fayos et al., 2007).

   Our analysis of the structural components of connectivity suggest that rill networks are key elements for the transfer of
water runoff and sediments. The densely developed rill network of Slope 1 acts as a primary factor enhancing the spatial
contiguity between the barely covered, source areas distributed along the hillslope and the outlet of the experimental slope,
largely increasing the structural connectivity of the system (Fig. 2c-d). In fact, rill networks provide very efficient erosive flow
routing pathways that largely facilitate the transmission of water and sediment fluxes across sections of the hillslopes with





little or no potential for runoff re-infiltration and sediment deposition (Nicolau, 2002; Bracken and Crocke, 2007; Moreno-de-las-Heras et al., 2010; Wester et al., 2014; Lu et al., 2019).

## 4.2 Functional connectivity: formation of connected runoff and sediment fluxes

Rainfall characteristics and antecedent conditions dynamically interact with the range of structural elements of the hillslopes

to enable or enhance connected flow. Our analysis reveals that maximum rainfall intensity and antecedent precipitation are the most relevant storm event factors providing dynamic control of surface-patch to hillslope scale continuity (or functional connectivity) of runoff (Fig. 4). Both peak rainfall intensity and antecedent moisture conditions are commonly perceived as the main rainfall factors involved in the generation of runoff in Mediterranean landscapes (Calvo-Cases et al., 2003; Castillo et al., 2003; Cammeraat, 2004; Cantón et al., 2011; Mayor et al., 2011; Marchamalo et al., 2016; Marínez-Murillo et al., 2018;

Rodríguez-Caballero et al., 2014). Infiltration-excess runoff triggered by high intensity rainfall typically dominates the hydrological responses of Mediterranean hillslopes under dry conditions. Saturation-excess runoff may also occur in Mediterranean-dry hillslopes, particularly on soils previously wetted by antecedent rainfall, inducing saturation of the top layer of the soil profile with moderate intensity precipitation (Martínez-Mena et al. 1998; Puigdefábregas et al., 1999; Calvo-Cases et al., 2003; Castillo et al., 2003). The poor soil development conditions that characterize our reclaimed study sites may

facilitate these two runoff generation mechanisms (Nicolau and Asensio, 2000; Moreno-de-las-Heras, 2009). Whilst rapid formation of surface crusts in barely covered patches of these reclaimed soils can largely facilitate the formation of infiltration-excess runoff, the massive structure of the soils, particularly in intermediate to deep layers showing moderate to low vegetation root activity, can also facilitate the formation of runoff from the temporary saturation of the top (5-20 cm) soil layer (Nicolau, 2002; Moreno-de-las-Heras et al., 2011a).

Our results also indicate a higher efficiency of antecedent precipitation than peak rainfall intensity in providing conditions for the generation of patch- to hillslope-scale runoff continuity (Figs. 4b and c). Similarly, other Mediterranean-dry hillslope and catchment studies have highlighted the primary role of antecedent precipitation on establishing spatial continuity in the generation and routing of runoff (Puigdefábregas et al., 1999; Fitzjohn et al., 1998; Boix-Fayos et al., 2007; Marchamalo et al., 2016). Under dry antecedent conditions, runoff generation is spatially heterogeneous due to the fine-scale spatial variation

of infiltration capacity. This variability is generated by both patchy vegetation and soil variability, which promote large discontinuities in hydrological pathways inducing spatial isolation of runoff producing areas (Calvo-Cases et al., 2003; Boix-Fayos et al., 2007). Differently, wet conditions reduce soil infiltration capacity (Cerdà, 1997; Moreno-de-las-Heras et al., 2009) and blur the spatial variation in hydrological properties by facilitating the formation of runoff from saturation of the top and subsoil layers (Puigdefábregas et al., 1999; Calvo-Cases et al., 2003), thus resulting in increased spatial continuity of active

hydrological pathways for the generation and transference of runoff among scales and elements of the landscapes (Fitzjohn et al., 1999; Boix-Fayos et al., 2007; Marchamalo et al., 2016). Our results suggest that 30-50 mm of precipitation accumulated over an antecedent period of 10 days can notably enhance the hydrological connectivity in our study slopes (Fig. 4c). These conditions mainly take place in the area during the autumn and spring seasons.





The structural connectivity of the studied slope systems largely controls the functional responses of runoff connectivity to
rainfall intensity and antecedent precipitation (Figs. 4b and c). Differently to Slopes 2 and 3, where the low spatial contiguity
of runoff source areas strongly limits the impact of rainfall conditions on runoff connectivity, the surface-patch to hillslope-
scale continuity of runoff is highly sensitive to rainfall intensity and antecedent precipitation in Slope 1. The presence of a
well-developed rill network in this slope strongly increases the structural connectivity of the system and provides a very
efficient pathway for the routing of runoff. Modelling results by Reaney et al. (2014) explain the high efficiency of rill networks
in the transmission of runoff as a function of their effects on transfer distances. Overall, rills operate as channels that reduce
transfer distances in relation to effective contributing area, resulting in enhanced runoff transmission along the hillslope by
increased flow velocity and reduced length to flow concentration. In this study, the high runoff transmission efficiency of the
rill networks can be illustrated by the resulting cross-scale runoff continuity responses observed in the experimental slopes
during the most extremely connected events (top $C_R$ whisker values in Fig. 3c), recorded on early June and November 2008
for rather large rainfall events with moderate peak intensities occurring under wet antecedent conditions (~50 mm depth, 7-9
mm h$^{-1}$ I$_{30}$ and 35-45 mm API). In such conditions, the high runoff transmission efficiency of the rill network in Slope 1
resulted in the complete transference of patch-scale generated runoff to the outlet of the system, largely differing from the
poorly established hydrological connectivity of the two non-rilled slopes, where 60-70% path-scale runoff re-infiltrated
downslope before reaching the outlet.

Besides the key influence of rills for the spatial transmission of runoff, these hillslope structural elements were also found
to play a fundamental role in the generation of sediment fluxes and its spatial distribution. Overall, our experimental slopes
display two contrasting sedimentological behaviours that can be compared in light of our sediment connectivity results (Figs.
3b and d). While the two non-rilled slopes (Slopes 2 and 3) show, for all the events analysed, poorly connected sediment flows
characterized by important sediment deposition between the surface-patch and hillslope scales, sediment yield can very largely
increase (i.e., up to 5 times) with scale for the rilled system (Slope 1) when active rill incision takes place. Similarly, other
studies carried out in Mediterranean landscapes indicate that sediment yield generally decreases from the small-plot to the
hillslope scales in the absence of rills due to the loss of runoff by downslope re-infiltration, while soil loss per unit area
generally increases with plot length when rill erosion processes prevail (Boix-Fayos et al., 2006; Bargarello and Ferro, 2010;
Moreno-de-las-Heras et al., 2010; Cantón et al., 2011; Bargarello et al., 2018). In fact, runoff convergence in the rill networks
provides these erosive elements with the capacity to produce very important amounts of sediments, frequently resulting in
significant increases in sediment yield with slope length (Gimenez and Govers, 2001; Govers et al., 2007; Moreno-de-las-
Heras et al., 2011b; Wester et al., 2014; Lu et al., 2019). In this context, rills not only facilitate within-slope transference of
runoff and sediment fluxes but also work as powerful sources of sediments that can significantly contribute with freshly eroded
particles to the analysed sediment fluxes between the surface-patch and hillslope scales.

Rainfall intensity emerges as the main storm property controlling the spatial continuity and scaling of sediment fluxes in
our study slopes, particularly for the rilled system (Slope 1), where the ratio of hillslope to surface-patch sediment production
(or functional connectivity of sediments) strongly increases with mean rainfall intensity (Fig. 5b). In Mediterranean hillslopes,



storm intensity provides direct control on splash erosion and largely influences –through runoff production and concentration– the processes of sheet wash, sediment transport and both rill and gully incision (Bargarello and Ferro, 2010; Moreno-de-las-Heras et al., 2010; Mayor et al., 2011; Cantón et al., 2011; Gallart et al., 2013). These links are frequently identified in the form of strong linear correlations between sediment yield and peak rainfall intensity (e.g., Cammeraat, 2004; Rodríguez-Caballero et al., 2014). In the present study, sediment connectivity is better explained by mean rainfall intensity, which may suggest enhanced conditions for sediment transfer and rill incision by sustained (rather than peak) high intensity rainfall. However, the high correlation that links the maximum ($I_{30}$) and mean ($I_m$) rainfall intensities of the analysed events (Pearson's R= 0.92, p<0.01) reveals that the storms displaying the best conditions for the formation of spatially connected sediment flows along the study period (top $C_S$ whisker values in Fig. 3d) were characterized by both high peak and averaged rainfall intensity (up to 33 and 6 mm h$^{-1}$ $I_{30}$ and $I_m$, respectively). These enhanced conditions for the production and routing of sediments were recorded during the summer season, in the form of high intensity convective rainfall. Previous erosion research in this experimental site has highlighted the erosive capacity of late spring and summer convective storms, which are responsible for up to 80% annual soil loss in the area (Nicolau, 1992; Moreno-de-las-Heras et al., 2010; Merino-Martín et al., 2012a).

The frequently observed presence of thresholds and the non-linear character of landscape hydro-geomorphological responses are to a large extent related to the runoff and sediment connectivity that are responsible for transferring surface fluxes of resources from the small plot and surface-patch scales to broader hillslope and catchment scales (Puigdefábregas et al., 1999; Cammeraat, 2004; Bracken and Croke, 2007; Wainwright et al., 2011; Moreno de las Heras et al., 2012; Sidle, 2017). Evidence of these dynamics is provided in this work by the non-linear relationships that link within-slope (functional) connectivity of runoff and sediment fluxes with the runoff coefficients and soil losses observed in the studied systems at the hillslope scale (Figs. 3e and f). Particularly, critical loss of the capacity for redistributing surface fluxes by runoff re-infiltration and sediment deposition mechanisms in the analysed slope systems results in very large increases in hillslope-scale runoff production and soil loss. These non-linear hydro-geomorphological responses are strongly conditioned by the dynamics of Slope 1, where the presence of a well-organized rill network provides the system with cross-scale hillslope structural elements for intensive sediment production and effective flow routing.

Substantial increases in functional connectivity can dynamically feed back into the structural aspects of connectivity by modifying the spatial organization of preferential flowpaths for the production and transmission of runoff and sediments (Turnbull et al., 2008; Wainwright et al., 2011; Okin et al., 2015; Turnbull and Wainwright, 2019). Structural connectivity is perceived in our study as a static property of the explored systems during the entire period of analysis (October 2007 to December 2008), which took place after 20 years of landscape evolution from the initial slope reclamation stage. Changes in structural connectivity of these Mediterranean-dry human-made systems can be particularly important during earlier stages (first 5-10 years) of landscape evolution, when the spatial arrangement and redistribution of runoff and sediment fluxes largely shape the initial establishment and further dynamics of both vegetation patterns and rill networks (Nicolau and Asensio, 2000; Moreno-de-las-Heras et al., 2011b). In fact, the rill network of Slope 1 represents a key factor determining the high structural connectivity of this system, but its own existence can be attributed to the development of high levels of (runoff and sediment)



functional connectivity during the early stages of landscape evolution. Spatially explicit modelling frameworks of co-evolving landforms and vegetation patterns (e.g., Saco et al., under review) may facilitate further exploration of the long-term dynamic feedbacks that link functional and structural connectivity.

## 5 Conclusions

Connectivity has emerged in hydrological sciences as a powerful theoretical concept that can facilitate deep understanding of the movement of runoff and water-borne sediments between landscape locations and across spatial scales. In this study, we carry out a practical application of the conceptual elements of structural and functional connectivity for the analysis of surface-patch to hillslope-scale transmission of runoff and sediment fluxes in three Mediterranean-dry reclaimed mining slope systems showing different levels of long-term development of vegetation and rill networks.

Flowpath distribution analysis of runoff/sediment source areas was used as a metric of structural connectivity for the study slopes. The results from this metric revealed a key role of the hillslope position of vegetation and, more critically, the presence of rill networks controlling the spatial organization of preferential pathways for the production and routing of runoff and sediment fluxes. The interactions between the structural connectivity of the experimental slopes and both antecedent precipitation and rainfall intensity largely controlled functional connectivity, assessed in this study through the spatial continuity of runoff and sediment fluxes between the surface-patch and hillslope scales for 21 monitored, hydrologically active rainfall events. Both runoff and sediments were largely redistributed within the analysed slope systems in the absence of rill networks. The results showed that rainfall intensity and, more importantly, antecedent precipitation largely increased the spatial continuity of runoff fluxes under rilled slope conditions. Furthermore, rainfall intensity enhanced the (functional) connectivity of sediments in the analysed systems. These enhanced conditions for the spatial continuity of sediment fluxes were particularly critical in the presence of rill networks, where active rill incision under high intensity rainfall induced large non-linear increases in hillslope-scale sediment yield.

In sum, this study provides empirical evidence of the feasibility of using the connectivity concept for practical applications, remarking specifically its usefulness for understanding how hillslope structural elements dynamically interact with storm characteristics and rainfall conditions to generate spatially continuous runoff and sediment fluxes. Overall, our study approach of structural and functional connectivity offers a useful framework for assessing the complex links and controlling factors that regulate the generation and movement of runoff and sediments across different scales and elements of the landscape in Mediterranean-dry and other water-limited environments.

*Supplement link.* This online supplement contains complementary information on general site characteristics (Table S1), the rainfall and hydrological data records of the analysed events (Table S2), the full GLM configurations applied for the analysis of runoff and sediment connectivity (Tables S3 and S4, respectively), and supplementary details on the parameterization of the antecedent precipitation index (API) applied in this study (Supplementary Methods S1).



*Data availability.* The full hydrological data applied in this study can be found in Table S2 of the online supplement. The rest of the data is available upon request from MMdlH.

*Author contributions.* All the authors participated in the design of the study. LMM obtained the field data, with contributions from MMdlH, JMN and TE. Data pre-processing and analysis was performed by MMdlH and LMM. MMdlH led the writing of the paper, with significant
contributions from all the co-authors.

*Competing interests.* The authors declare that they have no conflict of interest.

*Acknowledgements.* This study was supported by a Juan de la Cierva fellowship (IJCI-2015-26463) funded by the State Research Agency of
535 the Spanish Ministry of Science, Innovation and Universities (MCIU), and a research project (DP140104178) funded by the Australian Research Council. MMdlH acknowledges support from the University of Newcastle (Australia) through an International Research Visit fellowship developed in summer 2017 that facilitated initial discussion and further development of this study. Field collection of the runoff and sediment yield data used in this study was supported by a PhD scholarship awarded to LMM by the University of Alcalá and by the projects CGL2010-21754-C02-02 and S2009AMB-1783, funded by the MCIU and the Regional Government of Madrid, respectively. We
are grateful to the Spanish National Geographical Institute, and particularly to Juan M. Rodriguez, for granting us access to the PNOA aerial images for this study. We also thank José A. Merino for his help in the development of the field tachometry campaigns and DEM associated products, and Jesús Romero for language corrections.

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





**Table 1.** Spearman's R correlations between the event characteristics/conditions (storm depth, rainfall duration, 15-min and 30-min maximum rainfall intensity, mean rainfall intensity, and antecedent precipitation) and the connectivity of runoff and sediments for the three experimental slopes.

| | | Connectivity of Runoff - $C_R$ | Connectivity of Sediments - $C_S$ |
|---|---|---|---|
| | Slope 1 | **0.50**\* | 0.21ns |
| Dp, storm depth | Slope 2 | 0.48\* | 0.19ns |
| | Slope 3 | **0.60**\*\* | 0.36ns |
| | Slope 1 | 0.23ns | -0.23ns |
| Rd, rainfall duration | Slope 2 | 0.18ns | -0.25ns |
| | Slope 3 | 0.45\* | 0.23ns |
| $I_{15}$, 15-minunte | Slope 1 | **0.53**\*\* | **0.78**\*\*\* |
| maximum rainfall | Slope 2 | **0.56**\*\* | **0.55**\*\* |
| intensity | Slope 3 | **0.51**\* | 0.39ns |
| $I_{30}$, 30-minunte | Slope 1 | **0.57**\*\* | **0.73**\*\*\* |
| maximum rainfall | Slope 2 | **0.61**\*\* | **0.64**\*\* |
| intensity | Slope 3 | **0.57**\*\* | 0.43\* |
| Im, mean rainfall | Slope 1 | **0.54**\* | **0.78**\*\*\* |
| intensity | Slope 2 | **0.65**\*\* | **0.67**\*\*\* |
| | Slope 3 | **0.50**\* | **0.58**\*\* |
| API, antecedent | Slope 1 | **0.79**\*\*\* | 0.47\* |
| precipitation index | Slope 2 | **0.75**\*\*\* | 0.46\* |
| (10 days, k=0.98) | Slope 3 | **0.75**\*\*\* | **0.64**\*\* |

Sig. codes: '\*\*\*' p<0.001; '\*\*' p<0.01; '\*' p<0.05; 'ns' not significant at α=0.05.
Spearman's R correlation values in bold are ≥0.50.



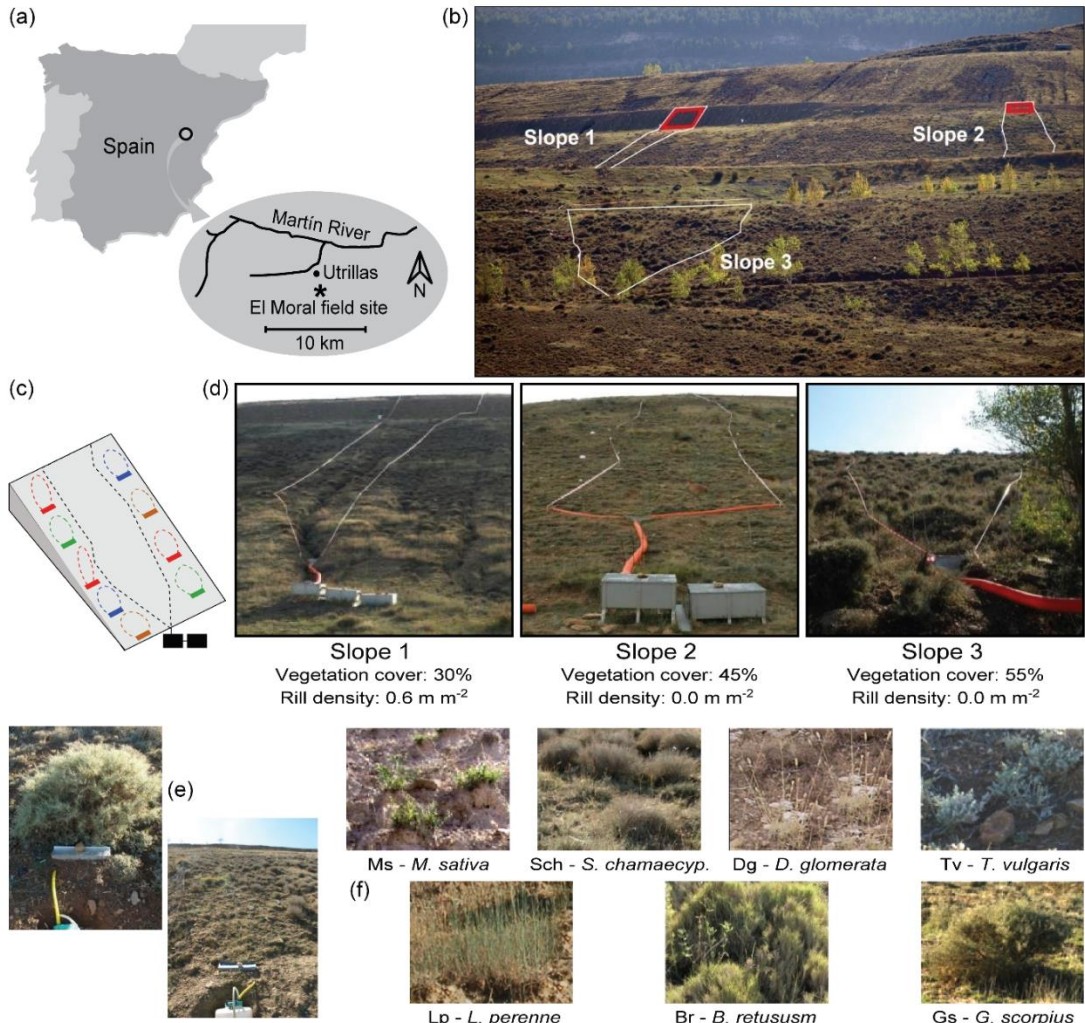

**Figure 1.** The Utrillas field site: (a) location map; (b) frontal view of the three experimental slopes (40° steep, berm sections located at the top of Slopes 1 and 2 are highlighted in red); (c) schematic representation of the experimental layout in the slopes (a hierarchical, scaled approach with patch- and hillslope-scale runoff/erosion plots, all naturally delimited); (d) frontal view of the hillslope-scale plots (catchment plots) and their general characteristics (vegetation cover and rill density); (e) two examples of patch-size plots (Gerlach troughs) in two different surface types (left picture-Gs surface type, right picture-Dg surface type); (f) detailed view of the seven surface types (vegetation communities) identified in the experimental slopes by Merino-Martin et al. (2012a). Names of the dominant plant species for the surface types: Ms, *Medicago sativa*; Sch, *Santolina chamaecyparissus*; Dg, *Dactilis glomerata*; Tv, *Thymus vulgaris*; Lp, *Lolium perenne*; Br, *Brachypodium retusum*; Gs, *Genista scorpius*.



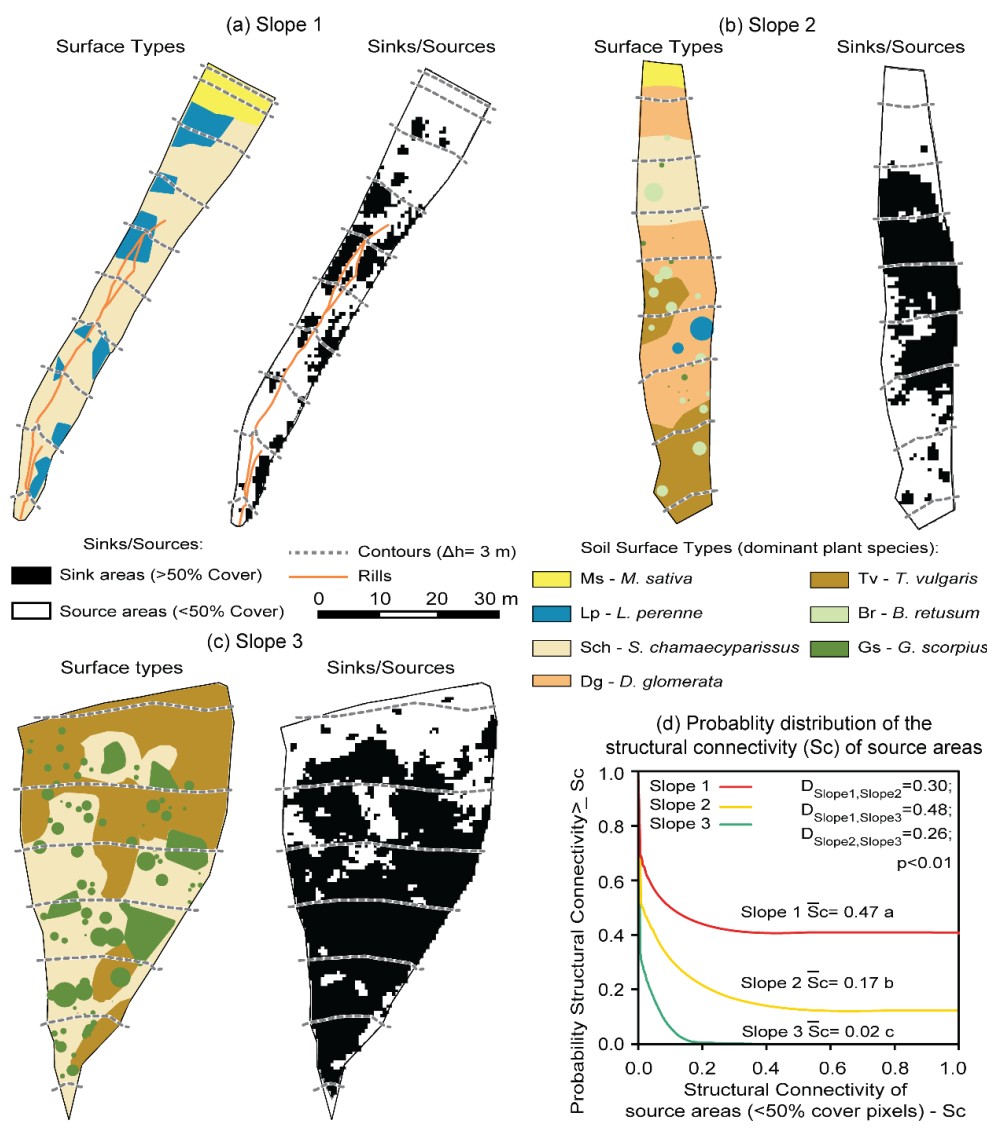

**Figure 2.** Structural connectivity: (a-c) spatial organization of surface types (vegetation communities) and sink/source areas (i.e., areas with above/below 50% vegetation cover) in the experimental slopes; (d) cumulative probability distribution function (CDF) of the structural connectivity (Sc) metric of source areas for the experimental slopes (mean structural connectivity values, $\overline{Sc}$, are provided for each slope system). Surface type maps and digital elevation data were derived from a field tachometry campaign (Merino-Martin et al., 2015). Binary maps (0.5 m pixel resolution) of sinks and sources are derived from a multispectral aerial picture (Spanish National Plan for Aerial Orthophotography, PNOA). Different letters in $\overline{Sc}$ values displayed on graph (d) indicate significant differences at α=0.05. Tested using two-sample Kolmogorov-Smirnov tests (D statistics and p levels are shown in the graph).



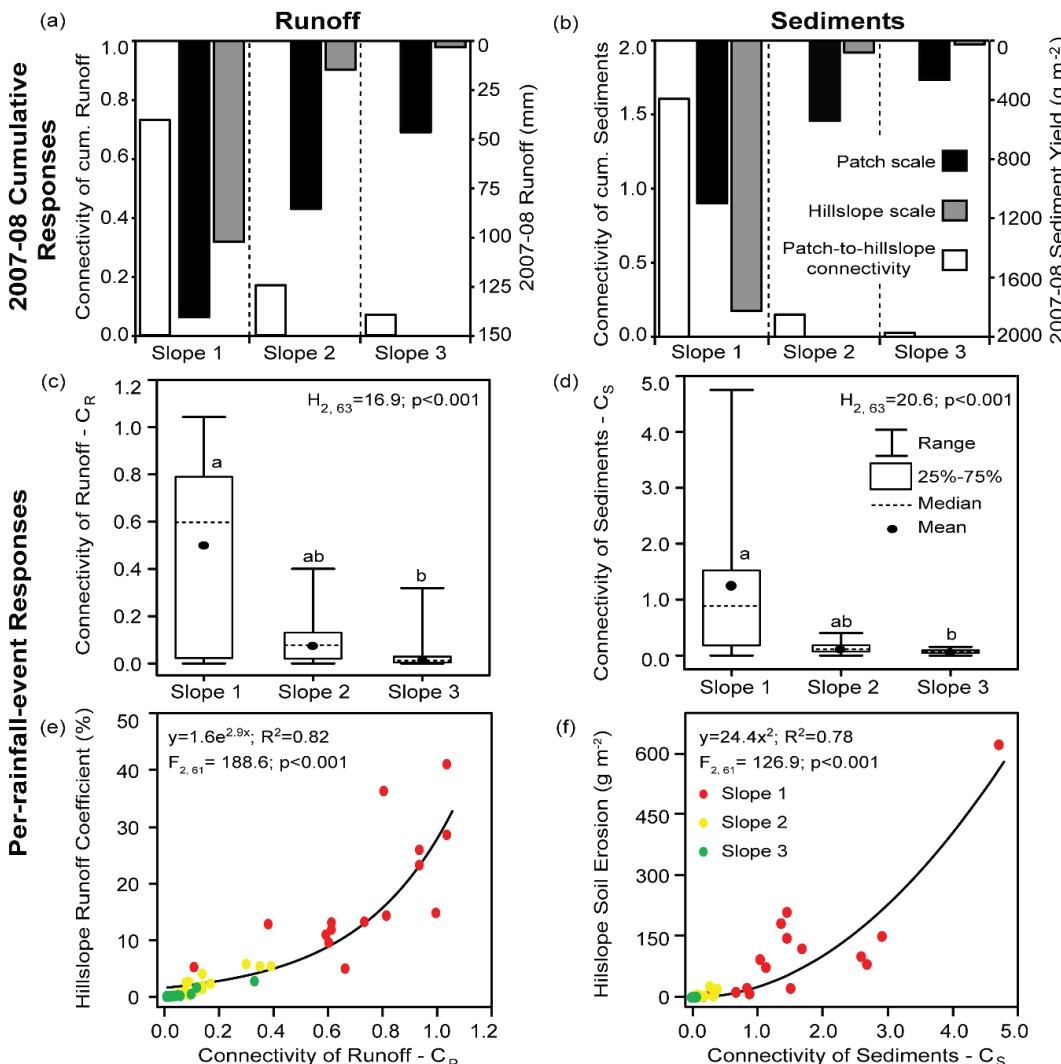

**Figure 3.** Functional connectivity: (a-b) 2007-08 cumulative and (c-d) per-rainfall-event connectivity of (a, c) runoff ($C_R$) and (b, d) sediments ($C_S$) in the experimental slopes; (e-f) relationship between the (per-rainfall-event) connectivity of runoff and sediments and hillslope-scale runoff coefficient and soil erosion. Connectivity of runoff/sediments across scales is represented as the ratio of hillslope to patch-scale runoff/sediments (white bars). Runoff and sediment yield at both the hillslope scale (grey bars) and the (integrated, proportionally weighed) patch scale (black bars) are provided for the 2007-08 cumulative metrics (graphs (a) and (b), secondary vertical axis). Different letters in bars of graphs (c) and (d) indicate significant differences for the three slopes at $\alpha=0.05$. Tested using Kruskal-Wallis ANOVA (H statistics and p levels are shown in the graphs).

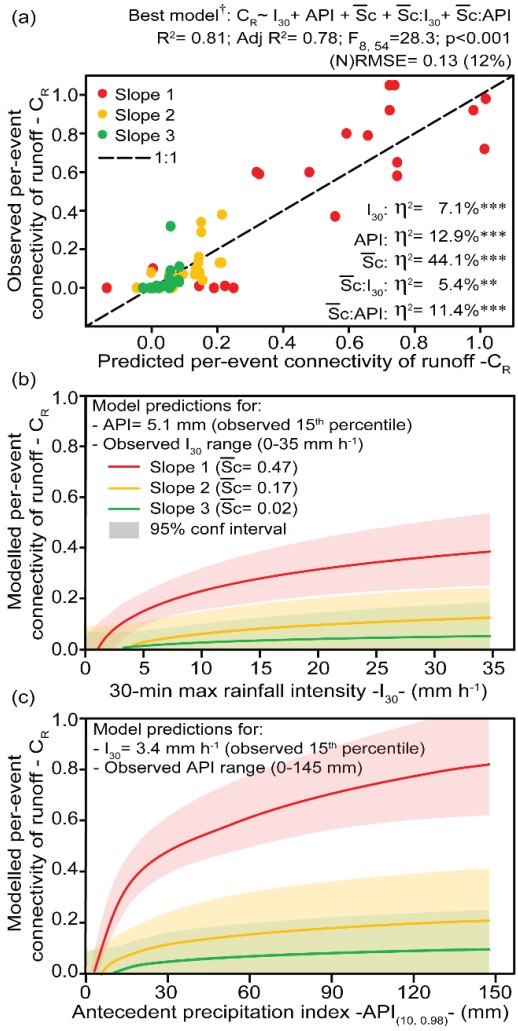

**Figure 4.** Structural and dynamic control of patch- to hillslope-scale runoff connectivity: (a) best-supported model of runoff connectivity and corresponding observed *versus* predicted runoff connectivity values (the structure, $R^2$, Adj $R^2$, F statistic, p value and root-mean-square error of the model are detailed on top of the graph; $\eta^2$ values within the graph indicate the % variance explained by the direct and interaction terms of the model); (b) modelled $I_{30}$ and $\overline{S}c$ effects on runoff connectivity (API is fixed at 5.1 mm, 15th percentile of observed API values); (c) modelled API and $\overline{S}c$ effects on runoff connectivity ($I_{30}$ is fixed at 3.4 mm h$^{-1}$, 15th percentile of observed values). Abbreviations: $C_R$, per-rainfall-event functional connectivity of runoff; $\overline{S}c$, mean structural connectivity of "source areas" in the experimental slopes; $I_{30}$, 30-min maximum rainfall intensity; API, antecedent precipitation index; ':', interactions between $\overline{S}c$ and the co-variables; (N)RMSE, (normalized) root-mean-square error; $\eta^2$, eta-squared statistic (effect size). Sig. codes: '***' $p<0.001$; '**' $p<0.01$; '*' $p<0.05$; 'ns' not significant at $\alpha=0.05$. Notes: †, the model takes log-transformed values for the co-variables $I_{30}$ and API; the graphs (b) and (c) show back-transformed $I_{30}$ and API values for the modelled relationships.





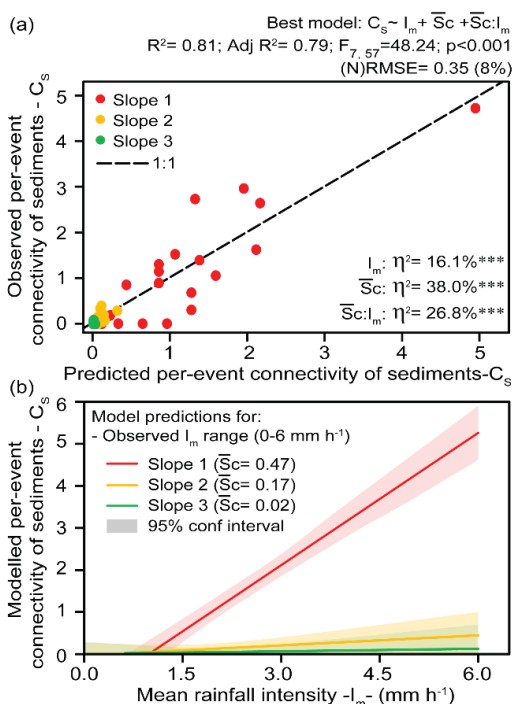

**Figure 5.** Structural and dynamic control of patch to hillslope-scale sediment connectivity: (a) best-supported model of sediment connectivity and corresponding observed *versus* predicted sediment connectivity values (the structure, $R^2$, Adj $R^2$, F statistic, p value and root-mean-square error of the model are detailed on top of the graph; $\eta^2$ values within the graph indicate the % variance explained by the direct and interaction terms of the model); (b) modelled Im and $\overline{S}c$ effects on runoff connectivity. Abbreviations: $C_S$, per-rainfall-event functional connectivity of sediments; $\overline{S}c$, mean structural connectivity of "source areas" in the experimental slopes; Im, mean rainfall intensity; ':', interactions between $\overline{S}c$ and the co-variables; (N)RMSE, (normalized) root-mean-square error; $\eta^2$, eta-squared statistic (effect size). Sig. codes: '***' $p<0.001$; '**' $p<0.01$; '*' $p<0.05$; 'ns' not significant at $\alpha=0.05$.