# Peer review of "Structural and functional control of surface-patch to hillslope-scale runoff and sediment connectivity in Mediterranean-dry reclaimed slope systems"

_Hydrology and Earth System Sciences, 2019_

## Referee Comment (RC1) · Anonymous Referee #1 · 31 Jan 2020

General comments:

In this paper presents a promising way to put the theoretical concept of structural and functional hydrological connectivity into practice by evaluating the connectivity between patch- and hillslope-scale with innovative measures for hydrological connectivity. Definition and measures of hydrological connectivity is an important field of hydrological research and offers additional value for sedimentological and geomorphological research. This study uses a threshold for vegetation cover combined with a high resolution digital elevation model to derive a measure for structural connectivity. Functional connectivity was determined for a defined precipitation event as the ratio of runoff/sediment contributions from the hillslope scale to the corresponding contributions on a virtual hillslope represented the integrated patch-scale contributions. Functional hydrological and sedimentological connectivity was successfully modeled using a generalized linear model. Model predictors included various measures of precipitation data as well as the structural connectivity measure. Surveyed data, methods and results contribute to the understanding of hydrological processes and the practical use of the hydrological connectivity concept in the Mediterranean-dry. Thus, I recommend the publication after the revision of this manuscript.

Specific comments:

Line 14: The first sentence is very general: "multiple factors", "variety of spatial scales", "variable degrees of connection". The sentence is also closely related to the second sentence. I suggest to merge the content in one precise sentence. You may also introduce the "Mediterranean-dry reclaimed mining slope systems" here to avoid confusion with the term "systems" later and also introduce an abbreviation for the full term for later in the text.

Line 15: Connection or connectivity?

Line 15: "In these systems" – there are no systems defined before.

Line 16: movement of water, runoff is already moving water.

Line 18: The sub-sentence beginning with "or the extent to which..." interrupts the reading flow, I suggest to transfer the sub-sentence into a second sentence.

Line 21: Same as line 18, better breaking the sentence into two parts, or leaving out the sub-sentence "determined as...". This leaves space to mention the GLM model in the abstract.

Line 21: "...was further explored..." may be changed to e.g. "...was calculated as...".

Line 22: The sentence may be rephrased like "Functional hydrological connectivity during precipitation events was found to be dynamically controlled by antecedent precipitation conditions and rainfall intensity and further strongly modulated by the structural connectivity of the slopes"

Line 24: "On slopes without rill networks, both runoff. . .

Line 25: "analyzed systems": there are no defined systems, may use e.g. hillslopes or research slopes

Line 29: transference of both "water" and sediment (without yield).

Line 34-40: These sentences are very close to the first sentences of the abstract. Rephrase either of them.

Line 36: Connection or connectivity? See also line 15. Please be specific about the terminology and definition of hydrological connectivity (also line 42).

Line 46: could be misread as "transfer of sediment fluxes". Better just "transfer of water and sediments" or "fluxes of water and sediment".

Line 47: I suggest to use: "the activation of connections of runoff. . ."

Line 47: I suggest not to write "In the case of runoff. . ." but "Functional connectivity of runoff depends on the dynamics. . .". Also I suggest to split this sentence to have one sentence for the runoff sub-sentence and one for the sedimentological.

Line 52: Leave out the "For example,"

Line 55: Leave out the "In fact,"

Line 56: In stead of "terraces) controls" may use " were shown to control" also: "from a structural connectivity perspective"

Line 63: This sentence may needs to be rephrased. The strength of the transport vector may be important for the sedimentological functional connectivity for pure hydrological

connectivity the establishment of a water flux between the patches of the landscape already represent fully functional connectivity between those patches no matter how big the flux is.

Line 65: "to determine the initiation of runoff and thus, the transport of water and sediments. . ."

Line 76: You could make use of an abbreviation from line 14 here.

Line 81: to my understanding the routing of runoff is part of the structural connectivity while the processes which cause infiltration/excess of water to initialize, maintain or interrupt the flow of water is part of the functional hydrological connectivity.

Line 84/88/91/95: Use abbreviation for the slope system.

Line 91: "transference of water. . ." Either Line 101 or as is in Line 129: Add a short sentence like: "The field work was accomplished between October 2007 and November 2008." After mentioning the dates of the survey no need for further repetition of the dates during the methods/results/discussion e.g. line 124/126. . ..

Line 103: Sentence is incomplete and does not make sense.

Line 105: Just: "Remarkable is. . ."

Line 115: This sentence suggests that Slope 2 also has significant amounts of overland flow and erosion, which to my understanding is not the case.

Line 121-125: This may also part of the results section.

Line 127: Already mentioned that in the abstract and introduction. No need to have that long introduction here for the methods.

Line 130: ". . .Merino-Martin et al., 2012a), that included naturally delimited runoff/erosion plots distributed at the (i) hillslope and the (ii) surface-patch scale."

Line 136-139: You mention Fig. 1d and 1f but not 1e. Usually the parts of the figures

are described according to their alphabetic order. Either restructure the text or the figure.

Line 140-143 & fig. 1f: Categories for the species would increase direct readability of the figure. E.g. Medicago sativa (Ms – A), Dactylis glomeratea (Dg - B), Santolina chamaecyparissus (Sch - B),...

Line 150-151: Sedimentological methods, may adjust header of the chapter to field measurements.

Line 153-169: Climatological, soil hydrological and statistical analytical methods mixed. I suggest to split the statistical part from the pure data acquisition part. A table showing an overview of the climatological statistics would be beneficial also for the introduction of the predictors for the GLM later on.

Line 171: "Previous research carried out..." (References missing!).

Line 172: Why using a range here when a non-dynamic threshold of 50% is applied?

Line 179: (0.5m resolution)

Line 184: "To this end," is a fill word and can be deleted.

Line 199: Maybe better: "...until a sink (i.e. >50% vegetation cover) or the outlet of the system is reached." And "outlet of the system is reached" is unclear which system patch or hillslope? In general, introducing a figure to illustrate the different steps of the calculation and also the use of mathematical symbols and equations to clarify the calculated ratio in line 201 could help to increase understanding for the reader.

Line 206: "Mean Sc values ((Sc) ÌĚ)...": The mean should be indicated by a dash above the whole symbol of which the mean is calculated.

Line 209-214: Again a repetition of the introduction sentences. It would be sufficient to leave it to very short general introduction sentences for the sub-headers in the methods.

Line 213: I recommend to stick with functional connectivity to stay within the framework of hydrological connectivity and not switch between spatial continuity and functional connectivity. Within the methods functional connectivity should defined for this study.

Line 214-220: Same as for structural connectivity: Figure and mathematical structure could help to increase understanding. Both connectivity measures could be visualized next to each other in one figure.

Line 233-238: This may not belong in the section under the header functional connectivity but in a section of statistical analysis.

Line 234: No need to mention the dates of the study period again.

Line 239: Same header as previous sub-section. Needs to state the statistical modeling/analysis.

Line 240-248: Have you checked for correlations among the predictors. The predictor set used in a GLM should not have high correlated predictors.

Line 240: As stated above (line 153-169) a table for the predictors would be helpful. This could be referenced here instead of "(Dp, Rd, I15, I30, and Im)"

Line 243: I suggest: "We modeled Cr and Cs using a generalized linear model (GLM, Christensen, 2002) approach with an automated stepwise backward model selection." Which link function did you use? Which program was used to implement the model and model selection?

Line 259: "This fact violates..." the relation of the "this fact" is unclear. I guess you mean the values of >1 for Cs. But the sentence ends with the values <1 for Cr which is not a violation against the GLM assumptions.

Line 260: either reference the suggested table again or may write "...transformation to the climatological co-variables..."

Line 263/265: no need to mention the vegetation cover again as it is defined in the

methods.

Line 269/276/279: "not physically linked"/ "physical contiguity" you mean "were not connected"/ "structural (functional) connectivity"?

Line 271: "...largely interfering the structural connection..."

Line 277: "almost 50%"?

Line 278: "...lower (12% for Slope 2 and <1% for Slope 3)..."

Line 283: "Functional connectivity of runoff across scales showed important differences..." The information in between was previous mentioned in the methods.

Line 285: "... decreased from the [...] hillslope-scale from Slope 1 to Slope 3 (Figure 3a)".

Line 288: "that 72% of..."

Line 289: "...of the system and 28% of the runoff was redistributed or re-infiltrated."

Line 290: Please just use the precise numbers here and not "less than" etc.

Line 305: "(Cs=0)"

Line 306: "...other events at Slope 1 hillslope-scale..."

Line 369: could be use of the abbreviation for Mediterranean-dry r.s.s.

Line 382: The large differences between Slope 2 and 3 were not pointed out in the results.

Line 424: "... largely controls the functional connectivity of the runoff responses..."

Line 461-463: The differences might also be explained by differences in temporal resolution of the precipitation measurements.

Line 497: "...movement of water..."

Line 502: This sentence has a very complicated structure. I suggest to re-write the sentence and may break it into two. You also mention rills as "preferential pathways" in your conclusions. This topic could be a little bit more emphasized in the discussion as it is a dominant element for the generation of runoff and sediment fluxes.

Technical corrections:

Line 43: "were proposed" instead of "have been proposed"

Line 68: "Several research approaches were applied. . ."

Line 696: Year missing. Link of public access of the review available?

Fig. 4: versus in captions needs to not italic

Tables:

A table summarizing predictor variables for the GLM would be benefitial. Figures:

Fig. 1a: The local map could be enlarged compared to the overview map of Spain.

Fig. 1e: The setup is hard to see in the images. Taking the lower part of the left image may would be sufficient. Adding a schematic may would be helpful.

Fig. 1f: Colored classes or class indication with capital letters for the dominant species of the three hillslopes could help to connect the species to the related hillslopes. If colors are used they can be also used to indicate the corresponding slopes in Fig. 1b and d.

Fig. 2-5: Please adjust the color scheme to suit the needs of colorblind. Testing the figures can be done by e.g. https://www.color-blindness.com/coblis-color-blindness-simulator/

Fig. 4/5: Abbreviations do not necessarily be explained in the captions.

572, 2020.

---

## Author Comment (AC1) · 27 Mar 2020

**Response to the interactive comment of Anonymous Referee #1**

on "Structural and functional control of surface-patch to hillslope-scale runoff and sediment connectivity in Mediterranean-dry reclaimed slope systems" by M. Moreno de las Heras et al.

The comments of the reviewer are shown below in italics. Our responses are presented below each comment in regular font. Proposed changes in the text as a consequence of the adaptation of the paper to the referee's comments are presented between quotation marks and in italics in our responses.

**General comment:**

In this paper presents a promising way to put the theoretical concept of structural and functional hydrological connectivity into practice by evaluating the connectivity between patch- and hillslopescale with innovative measures for hydrological connectivity. Definition and measures of hydrological connectivity is an important field of hydrological research and offers additional value for sedimentological and geomorphological re-search. This study uses a threshold for vegetation cover combined with a high resolution digital elevation model to derive a measure for structural connectivity. Functional connectivity was determined for a defined precipitation event as the ratio of runoff/sediment contributions from the hillslope scale to the corresponding contributions on a virtual hillslope represented the integrated patch-scale contributions. Functional hydrological and sedimentological connectivity was successfully modeled using a generalized linear model. Model predictors included various measures of precipitation data as well as the structural connectivity measure. Surveyed data, methods and results contribute to the understanding of hydrological processes and the practical use of the hydrological connectivity concept in the Mediterranean-dry. Thus, I recommend the publication after the revision of this manuscript.

**Response to the general comment:** We thank Referee # 1 for his/her positive assessment of the scope and contents of our study, and for his/her thoughtful comments and detailed edits, which will help to improve significantly our paper. All his/her comments are addressed in detail below.

**Specific comments (SCs):**

**SC Line 14:** The first sentence is very general: "multiple factors", "variety of spatial scales", "variable degrees of connection". The sentence is also closely related to the second sentence. I suggest to merge the content in one precise sentence. You may also introduce the "Mediterranean-dry reclaimed mining slope systems" here to avoid confusion with the term "systems" later and also introduce an abbreviation for the full term for later in the text.

**Response to SC Line 14:** Following the recommendations, we will simplify the first three sentences of the abstract, removing imprecise concepts and introducing clearly the study systems: *"Connectivity has emerged as a useful concept for exploring the movement of water and sediments between landscape locations and across spatial scales. In this study, we examine the structural and functional controls of surface-patch to hillslope-scale runoff and sediment connectivity in three Mediterranean-dry reclaimed mining slope systems that have different long-term development levels of vegetation and rill networks". The use of an abbreviation for the term "Mediterranean-dry reclaimed mining slope systems" will be introduced a bit later in the text, in the Introduction section.*

**SC Line 15:** *Connection or connectivity?**

**Response to SC Line 15:** We meant "connectivity". This specific sentence will be removed in the revised version of the paper, following the previous recommendations of comment SC Line 14.

**SC Line 15(b):** *"In these systems" – there are no systems defined before.**

**Response to SC Line 15(b):** After merging and simplifying the first three sentences of the abstract (please, see above our response to SC Line 14 for details of the proposed changes) all vague citations to generic "systems" will be removed.

**SC Line 16:** *movement of water, runoff is already moving water.**

**Response to SC Line 16:** Following the recommendations, we will change "movement of runoff" to "movement of water".

**SC Line 18:** The sub-sentence beginning with "or the extent to which…" interrupts the reading flow, I suggest to transfer the sub-sentence into a second sentence.

**Response to SC Line 18:** Following the recommendations, we will transfer the sub-sentence to end of the text structure: "*Structural connectivity was assessed using flowpath analysis of coupled vegetation distribution and surface topography, providing field indicators of the extent to which surface patches that facilitate runoff/sediment production are physically linked to one another in the studied hillslopes*".

**SC Line 21:** Same as line 18, better breaking the sentence into two parts, or leaving out the subsentence "determined as...". This leaves space to mention the GLM model in the abstract.

**Response to SC Line 21:** Thanks for the suggestions. We will remove the sub-sentence: *"Functional connectivity was calculated using the ratio of surface-patch to hillslope-scale observations of runoff and sediment yield for 21 monitored hydrologically active rainfall events"*. In addition, we will introduce new information mentioning our modelling methods in the abstract: "The impact of the dynamic interactions between rainfall conditions and structural connectivity on functional connectivity were further analyzed using general linear models with a backward model structure selection approach".

SC Line 21(b): "...was further explored..." may be changed to e.g. "...was calculated as...".

**Response to SC Line 21(b):** Following the recommendations, we will change "was further explored" to "was calculated".

**SC Line 22:** The sentence may be rephrased like "Functional hydrological connectivity during precipitation events was found to be dynamically controlled by antecedent precipitation conditions and rainfall intensity and further strongly modulated by the structural connectivity of the slopes"

**Response to SC Line 22:** We will rephrase the sentence following these specific recommendations.

SC Line 24: "On slopes without rill networks, both runoff..."

**Response to SC Line 24:** We will rephrase the sentence following these specific recommendations.

**SC Line 25:** *"analyzed systems": there are no defined systems, may use e.g. hillslopes or research slopes*

**Response to SC Line 25:** Following the recommendations, we will change "systems" to "hillslopes".

**SC Line 29:** *transference of both "water" and sediment (without yield).*

**Response to SC Line 29:** We will rephrase the sentence following these specific recommendations.

**SC Line 34-40:** These sentences are very close to the first sentences of the abstract. Rephrase either of them.

**Response to SC Line 34-40:** The text of the first sentences of the abstract will be modified very considerably (please, see above our response to SC Line 14 for details of the proposed changes), also minimizing redundancy with the information of the first sentences in the Introduction.

**SC Line 36:** *Connection or connectivity? See also line 15. Please be specific about the terminology and definition of hydrological connectivity (also line 42).*

**Response to SC Line 36:** Following the recommendations and in order to provide coherence in the use of terminology, we will change along the paper "connection" to "(hydrological) connectivity".

**SC Line 46:** could be misread as "transfer of sediment fluxes". Better just "transfer of water and sediments" or "fluxes of water and sediment".

**Response to SC Line 46:** Following the recommendations, we will change "transfer of water and sediment fluxes" to "transfer of water and sediments".

**SC Line 47:** *I suggest to use: "the activation of connections of runoff..."*

**Response to SC Line 47:** Following the recommendations, we will change "the generation of active connections" to "the activation of connections".

**SC Line 47b:** I suggest not to write "In the case of runoff..." but "Functional connectivity of runoff depends on the dynamics...". Also I suggest to split this sentence to have one sentence for the runoff sub-sentence and one for the sedimentological.

**Response to SC Line 47b:** We will add the suggested changes: *"Functional connectivity of runoff depends on the dynamics of overland flow generation, routing, and downward re-infiltration. For sediments, functional connectivity is a function of the detachment, entrainment, deposition and remobilization of sediments across scales"*.

**SC Line 52: Leave out the "For example,"**

**Response to SC Line 52:** Following the recommendations, we will remove "For example" in the text.

SC Line 55: Leave out the "In fact,"

**Response to SC Line 55:** Following the recommendations, we will remove "In fact" in the text.

**SC Line 56:** *Instead of "terraces) controls" may use "were shown to control" also: "from a structural connectivity perspective"*

**Response to SC Line 56:** We will add the suggested changes: "The spatial arrangement of surface features (e.g., vegetation cover, rills, gullies, channels, terraces) were shown to control the distribution of source and sink elements in these landscapes from a structural connectivity perspective, largely driving the production and transference of runoff and sediments across scales".

**SC Line 63:** This sentence may needs to be rephrased. The strength of the transport vector may be important for the sedimentological functional connectivity for pure hydrological connectivity the establishment of a water flux between the patches of the landscape already represent fully functional connectivity between those patches no matter how big the flux is.

**Response to SC Line 63:** Following the recommendations, we will rephrase the sentence: "The interactions between precipitation conditions and the structural connectivity of a landscape determines functional connectivity".

**SC Line 65:** "to determine the initiation of runoff and thus, the transport of water and sediments..."

**Response to SC Line 65:** We will rephrase the sentence following these specific recommendations.

**SC Line 76:** You could make use of an abbreviation from line 14 here.

**Response to SC Line 76:** The term "reclaimed mining slope systems" will be abbreviated along the paper from this line: "*Mediterranean-dry reclaimed mining slope systems (hereafter RMSSs)*".

**SC Line 81:** to my understanding the routing of runoff is part of the structural connectivity while the processes which cause infiltration/excess of water to initialize, maintain or interrupt the flow of water is part of the functional hydrological connectivity.

**Response to SC Line 81:** Following the recommendations and for coherence with the applied terminology, we will reword this sentence: *"the processes that initialize, maintain or interrupt the fluxes of water and sediments from the surface patch to the broader, hillslope scale"*.

**SC Line 84/88/91/95: Use abbreviation for the slope system.**

**Response to SC Line 84/88/91/95:** Following the recommendations, we will use the abbreviation "RMSS" in these lines and throughout the paper.

SC Line 91: "transference of water..."

**Response to SC Line 91:** Following the recommendations, we will change "transference of runoff" to "transference of water".

**SC Line (either) 101/129:** Add a short sentence like: "The field work was accomplished between October 2007 and November2008." After mentioning the dates of the survey no need for further repetition of the dates during the methods/results/discussion e.g. line 124/126....

**Response to SC Line (either) 101/129:** Following the recommendations, the fieldwork dates will be detailed in a single sentence located in the first lines of the Methods: *"The study site*

encompasses three experimental slopes, all north facing with a general gradient of about 20o (Figures 1b and 1d), which were surveyed intensively between October 2007 and December 2008".

**SC Line 103:** Sentence is incomplete and does not make sense.**

**Response to SC Line 103:** We will reword the sentence: "Mean annual precipitation (MAP) is 450 mm, most of which occurs in spring and autumn. Potential evapotranspiration (PET; Hargreaves and Samani, 1985) is around 900 mm and the hydrological deficit (MAP-PET) is approx. 450 mm, concentrated in the summer months (López-Martín et al., 2007)".

**SC Line 105: Just: "Remarkable is..."**

**Response to SC Line 105:** We will rephrase the sentence following these specific recommendations.

**SC Line 115:** This sentence suggests that Slope 2 also has significant amounts of overland flow and erosion, which to my understanding is not the case.**

**Response to SC Line 115:** The berm that act as a runoff contribution structure is much smaller in Slope 2 than in Slope 1, which limits its impact in the former as compared to the later. We will rephrase this sentence to avoid confusions: "*These variations occurred due to the existence of a very steep (40o) and bare soil, runoff contributing berm integrated at the top of two of the experimental slopes (Slopes 1 and 2, with berm sizes of 50 and 20 m2, respectively; Figure 1b). This runoff contributing structure promoted soil erosion and conditioned the early dynamics of the experimental slopes, particularly in Slope 1, where the berm area is bigger and produces important amounts of overland flow".*

**SC Line 121-125: This may also part of the results section.**

**Response to SC Line 121-125:** Although we understand the comment raised by the reviewer, we prefer to keep this information (gross runoff and sediment production of the slopes) in the description of the experimental site, as these numbers are already published in a previous paper (Merino-Martín et al. 2012a) and we do not apply any direct analysis of these gross variables (we just refer here to the published general information).

**SC Line 127:** Already mentioned that in the abstract and introduction. No need to have that long introduction here for the methods.

**Response to SC Line 127:** Following the recommendation, we will delete the redundant information from the text.

**SC Line 130:** *"…Merino-Martin et al., 2012a), that included naturally delimited runoff/erosion plots distributed at the (i) hillslope and the (ii) surface-patch scale."*

**Response to SC Line 130:** We will rephrase the sentence following these specific recommendations.

**SC Line 136-139:** You mention Fig. 1d and 1f but not 1e. Usually the parts of the figures are described according to their alphabetic order. Either restructure the text or the figure.

**Response to SC Line 136-139:** Please, note that panel Fig. 1e is already cited in the text (line 145), although it is cited after Fig. 1f. We will re-order the citations in the text and the panels in Fig. 1 to describe them according to their alphabetic order.

**SC Line 140-143 & fig. 1f:** Categories for the species would increase direct readability of the figure. E.g. Medicago sativa (Ms – A), Dactylis glomeratea (Dg - B), Santolina chamaecyparissus (Sch - B),...

**Response to SC Line 140-143 & Fig. 1f:** We believe that increasing the complexity of the species abbreviations in the text with new characters has probably little interest in terms of increased readability of the paper. However, we agree with Referee # 1 that linking these species with the hillslopes in the figure can be useful. We will modify panel (f) of figure 1 to link the dominant species with the hillslopes using the following labels/codes:

Ms - *M. sativa* (Slopes S1, S2)

Lp - *L. perenne* (Slopes S1, S2)

Sch - *S. chamaecyp.* Dg -(Slopes S1, S2, S3)

Br - *B. retususm* (Slope S2)

Tv - *T. vulgaris* (Slopes S2, S3)

Gs - *G. scorpius* (Slopes S2, S3)

**SC Line 150-151:** Sedimentological methods, may adjust header of the chapter to field measurements.

**Response to SC Line 150-151:** We will adjust the header of the (sub)section to: "2.2 Field acquisition methods of hydro-sedimentary and precipitation data".

**SC Line 153-169:** *Climatological, soil hydrological and statistical analytical methods mixed. I suggest* to split the statistical part from the pure data acquisition part. A table showing an overview of the climatological statistics would be beneficial also for the introduction of the predictors for the GLM later on.

**Response to SC Line 153-169:** Following the recommendations, we will split the statistical and data acquisition information and generate a new section in the Methods for the description of the statistical methods (section *"2.5 Data analysis and statistics"*). In addition, we will add the following new table that shows an overview of the precipitation condition variables applied in the study:

|                 | Description                    | Units              |
|-----------------|--------------------------------|--------------------|
| Dp              | Storm depth                    | mm                 |
| Rd              | Rainfall duration              | h                  |
| I 15 | 15-min max rainfall intensity  | mm h⁻¹             |
| I 30 | 30-min max rainfall intensity  | mm h -1 |
| Im              | ,
Mean rainfall intensity   | mm h -1 |
| API             | Antecedent precipitation index | mm                 |

**SC Line 171:** *"Previous research carried out..."* (*References missing!*).

**Response to SC Line 171:** Although the reference was already included in the original version of the paper (line 175: Moreno-de-las-Heras et al., 2009), we agree with the referee that the first line of the information is a better place to the citation in the text: "*Previous research carried out in the Utrillas field site applying small-scale (0.25 m2) rainfall simulations (Moreno-de-las-Heras et al., 2009) (...)".*

**SC Line 172:** Why using a range here when a non-dynamic threshold of 50% is applied?

**Response to SC Line 172:** We have modified the text, in line with the applied non-dynamic threshold of vegetation cover: *"surface patches with vegetation cover under 50% can generate important amounts of runoff/sediments"*.

SC Line 179: (0.5m resolution)

Response to SC Line 179: We will add the suggested change in the text.

**SC Line 184:** *"To this end," is a fill word and can be deleted.*

**Response to SC Line 184:** Following the recommendations, we will remove the expression "to this end".

**SC Line 199:** Maybe better: "...until a sink (i.e. >50% vegetation cover) or the outlet of the system is reached." And "outlet of the system is reached" is unclear which system patch or hillslope?

**Response to SC Line 199:** Following the recommendations, we will reword this sentence: "*until a sink (i.e., >50% vegetation cover) or the outlet of the hillslope is reached*".

**SC Line 199(b):** In general, introducing a figure to illustrate the different steps of the calculation and also the use of mathematical symbols and equations to clarify the calculated ratio in line 201 could help to increase understanding for the reader.

**Response to SC Line 199(b):** We agree with Referee # 1 that illustrating the different steps of the calculations in a figure can strongly facilitate understanding of the calculated index of structural connectivity. We will add the following schematic figure illustrated with a virtual example that will be linked to the text using fully explicit mathematical symbols and equations:

---

## Referee Comment (RC2) · Anonymous Referee #2 · 31 Mar 2020

Dear authors,

I find your manuscript very useful and a very good contribution for connectivity studies. I must say I found it very well and clear written. I can only say as a very minor correction that I would reduce the conclusion because I find the current one too long and descriptive. Congratulations!
* * *

---

## Author Comment (AC2) · 3 Apr 2020

**Response to the interactive comment of **Anonymous Referee # 2**

on "Structural and functional control of surface-patch to hillslope-scale runoff and sediment connectivity in Mediterranean-dry reclaimed slope systems" by M. Moreno de las Heras et al.

The comments of the reviewer are shown below in italics. Our responses are presented below each comment in regular font. Proposed changes in the text as a consequence of the adaptation of the paper to the referee's comments are presented between quotation marks and in italics in our responses.

**General comment:**

*Dear authors,*

*I find your manuscript very useful and a very good contribution for connectivity studies. I must say I found it very well and clear written. I can only say as a very minor correction that I would reduce the conclusion because I find the current one too long and descriptive.*

*Congratulations!*

**Response to the general comment:** We thank Referee # 2 for his/her very positive assessment of our study. Following the suggested changes for the conclusions, we will reduce significantly the length of this section (from 376 to 275 words), reducing also the descriptive character of the text:

''*We developed in this study a practical application of the conceptual elements of structural and functional connectivity for the analysis of surface-patch to hillslope-scale transmission of runoff and sediments in three Mediterranean-dry reclaimed mining slope systems showing different levels of long-term development of vegetation and rill networks. Our results revealed an important role of the hillslope position of vegetation patches on the distribution of potential runoff and sediment flowpaths. More critically, the rill networks emerged as key elements of structural connectivity in the slopes, providing preferential pathways that dominate the production, spatial organization and routing of the fluxes of water and sediments. On the other hand, both runoff and sediments were largely redistributed within the analysed slope systems in the absence of rill networks. The interactions between the structural connectivity of the experimental slopes and both antecedent precipitation and rainfall intensity largely controlled event functional connectivity. The results showed that rainfall intensity and, more importantly, antecedent precipitation largely increased the spatial continuity of runoff fluxes under rilled slope conditions, where active rill incision under high intensity rainfall induced large non-linear increases in hillslope-scale sediment yield.*

*In sum, this study provides empirical evidence of the feasibility of using the hydrological connectivity concept for practical applications, remarking specifically its usefulness for understanding how hillslope structural elements dynamically interact with storm characteristics and rainfall conditions to generate spatially continuous runoff and sediment fluxes. Overall, our study approach of structural and functional connectivity offers a useful framework for assessing the complex links and controlling factors that regulate the generation and movement of runoff and sediments across different scales and elements of the landscape in Mediterranean-dry and other water-limited environments*".

---

## Author Response (AR1)

Professor Erwin Zehe Executive Editor of HESS Handling editor of paper hess-2019-572

April 12th, 2020

Dear Prof. Erwin Zehe,

We are very pleased with the positive evaluation of our manuscript entitled "Structural and functional control of surface-patch to hillslope-scale runoff and sediment connectivity in Mediterranean-dry reclaimed slope systems" (ref. hess-2019-572). The article has clearly benefited from the helpful suggestions of the two anonymous referees.

Please, find below a complete list of answers to the comments and changes to the paper carried out to carefully address your remarks and the suggestions proposed by the two referees. The comments are shown below in italics. Our responses are presented below each comment in regular font. Changes in the text of the revised version of our paper are presented in our responses between quotation marks and in italics. A marked-up version of the manuscript showing the specific changes we made is submitted along with this letter.

Looking forward to hearing from you soon,

On behalf of the co-authors, Yours sincerely, Mariano Moreno de las Heras

Mariano Moreno de las Heras Institute of Environmental Assessment & Water Research Spanish Research Council (IDAEA-CSIC) C/ Jordi Girona 18-26, 08034 Barcelona Tel: +34 93 400 6100 mariano.moreno@idaea.csic.es

**1. Comlments by the Editor, Erwin Zehe**

**General comment:**

After a close look at your manuscript I concur with both reviewers, that your study needs minor revisions. In this respect I regard the outlined changes in your response to reviewer 1 as very much appropriate.

**Response to the general comment:** We thank Erwin Zehe for the positive evaluation of the paper. We have modified the paper according to the outlined changes that we proposed in the interactive discussion of the original paper. These changes and modifications are presented in detail below, along with our responses to the referee comments.

**2. Comments by Referee # 1**

**General comment:**

In this paper presents a promising way to put the theoretical concept of structural and functional hydrological connectivity into practice by evaluating the connectivity between patch- and hillslopescale with innovative measures for hydrological connectivity. Definition and measures of hydrological connectivity is an important field of hydrological research and offers additional value for sedimentological and geomorphological re-search. This study uses a threshold for vegetation cover combined with a high resolution digital elevation model to derive a measure for structural connectivity. Functional connectivity was determined for a defined precipitation event as the ratio of runoff/sediment contributions from the hillslope scale to the corresponding contributions on a virtual hillslope represented the integrated patch-scale contributions. Functional hydrological and sedimentological connectivity was successfully modeled using a generalized linear model. Model predictors included various measures of precipitation data as well as the structural connectivity measure. Surveyed data, methods and results contribute to the understanding of hydrological processes and the practical use of the hydrological connectivity concept in the Mediterranean-dry. Thus, I recommend the publication after the revision of this manuscript.

**Response to the general comment:** We thank Referee # 1 for his/her positive assessment of the scope and contents of our study, and for his/her thoughtful comments and detailed edits, which helped to improve significantly our paper. All his/her comments are addressed in detail below.

**Specific comments (SCs):**

**SC Line 14:** The first sentence is very general: "multiple factors", "variety of spatial scales", "variable degrees of connection". The sentence is also closely related to the second sentence. I suggest to merge the content in one precise sentence. You may also introduce the "Mediterranean-dry

reclaimed mining slope systems" here to avoid confusion with the term "systems" later and also introduce an abbreviation for the full term for later in the text.

**Response to SC Line 14:** Following the recommendations, we have simplified the first three sentences of the abstract, removing imprecise concepts and introducing clearly the study systems: "Connectivity has emerged as a useful concept for exploring the movement of water and sediments between landscape locations and across spatial scales. In this study, we examine the structural and functional controls of surface-patch to hillslope-scale runoff and sediment connectivity in three Mediterranean-dry reclaimed mining slope systems that have different long-term development levels of vegetation and rill networks". The use of an abbreviation for the term "Mediterranean-dry reclaimed mining slope systems" is introduced a bit later in the text, in the Introduction section.

**SC Line 15: Connection or connectivity?**

**Response to SC Line 15:** We meant "connectivity". This specific sentence was removed in the revised version of the paper, following the previous recommendations of comment SC Line 14.

**SC Line 15(b):** *"In these systems" – there are no systems defined before.**

**Response to SC Line 15(b):** After merging and simplifying the first three sentences of the abstract (please, see above our response to SC Line 14 for details of the changes) all vague citations to generic "systems" have been removed.

**SC Line 16:** *movement of water, runoff is already moving water.**

**Response to SC Line 16:** Following the recommendations, we have changed "movement of runoff" to "movement of water".

**SC Line 18:** The sub-sentence beginning with "or the extent to which..." interrupts the reading flow, I suggest to transfer the sub-sentence into a second sentence.

**Response to SC Line 18:** Following the recommendations, we have transferred the subsentence to end of the text structure: "*Structural connectivity was assessed using flowpath analysis* of coupled vegetation distribution and surface topography, providing field indicators of the extent to which surface patches that facilitate runoff/sediment production are physically linked to one another in the studied hillslopes".

**SC Line 21:** Same as line 18, better breaking the sentence into two parts, or leaving out the subsentence "determined as...". This leaves space to mention the GLM model in the abstract.

**Response to SC Line 21:** Thanks for the suggestions. We have removed the sub-sentence: *"Functional connectivity was calculated using the ratio of surface-patch to hillslope-scale observations of runoff and sediment yield for 21 monitored hydrologically active rainfall events"*. In addition, we have introduced new information mentioning our modelling methods in the abstract: "The impact of the dynamic interactions between rainfall conditions and structural connectivity on functional connectivity were further analyzed using general linear models with a backward model structure selection approach".

SC Line 21(b): "...was further explored..." may be changed to e.g. "...was calculated as...".

**Response to SC Line 21(b):** Following the recommendations, we have changed "was further explored" to "was calculated".

**SC Line 22:** The sentence may be rephrased like "Functional hydrological connectivity during precipitation events was found to be dynamically controlled by antecedent precipitation conditions and rainfall intensity and further strongly modulated by the structural connectivity of the slopes"

**Response to SC Line 22:** We have rephrased the sentence following these specific recommendations.

SC Line 24: "On slopes without rill networks, both runoff..."

**Response to SC Line 24:** We have rephrased the sentence following these specific recommendations.

**SC Line 25:** *"analyzed systems": there are no defined systems, may use e.g. hillslopes or research slopes*

**Response to SC Line 25:** Following the recommendations, we have changed "*systems*" to "*hillslopes*".

**SC Line 29:** transference of both "water" and sediment (without yield).

**Response to SC Line 29:** We have rephrased the sentence following these specific recommendations.

**SC Line 34-40:** These sentences are very close to the first sentences of the abstract. Rephrase either of them.

**Response to SC Line 34-40:** The text of the first sentences of the abstract have been modified very considerably (please, see above our response to SC Line 14 for details of the changes), also minimizing redundancy with the information of the first sentences in the Introduction.

**SC Line 36:** Connection or connectivity? See also line 15. Please be specific about the terminology and definition of hydrological connectivity (also line 42).

4

**Response to SC Line 36:** Following the recommendations and in order to provide coherence in the use of terminology, we have changed along the paper "connection" to "(hydrological) connectivity".

**SC Line 46:** could be misread as "transfer of sediment fluxes". Better just "transfer of water and sediments" or "fluxes of water and sediment".

**Response to SC Line 46:** Following the recommendations, we have changed "transfer of water and sediment fluxes" to "transfer of water and sediments".

**SC Line 47:** *I suggest to use: "the activation of connections of runoff..."*

**Response to SC Line 47:** Following the recommendations, we have changed "the generation of active connections" to "the activation of connections".

**SC Line 47b:** I suggest not to write "In the case of runoff..." but "Functional connectivity of runoff depends on the dynamics...". Also I suggest to split this sentence to have one sentence for the runoff sub-sentence and one for the sedimentological.

**Response to SC Line 47b:** We have added the suggested changes: "Functional connectivity of runoff depends on the dynamics of overland flow generation, routing, and downward reinfiltration. For sediments, functional connectivity is a function of the detachment, entrainment, deposition and remobilization of sediments across scales".

**SC Line 52: Leave out the "For example,"**

**Response to SC Line 52:** Following the recommendations, we have removed "For example" in the text.

SC Line 55: Leave out the "In fact,"

**Response to SC Line 55:** Following the recommendations, we have removed "In fact" in the text.

**SC Line 56:** *Instead of "terraces) controls" may use "were shown to control" also: "from a structural connectivity perspective"*

**Response to SC Line 56:** We have added the suggested changes: "The spatial arrangement of surface features (e.g., vegetation cover, rills, gullies, channels, terraces) were shown to control the distribution of source and sink elements in these landscapes from a structural connectivity perspective, largely driving the production and transference of runoff and sediments across scales".

**SC Line 63:** This sentence may needs to be rephrased. The strength of the transport vector may be important for the sedimentological functional connectivity for pure hydrological connectivity the establishment of a water flux between the patches of the landscape already represent fully functional connectivity between those patches no matter how big the flux is.

**Response to SC Line 63:** Following the recommendations, we have rephrased the sentence: "The interactions between precipitation conditions and the structural connectivity of a landscape determines functional connectivity".

**SC Line 65:** "to determine the initiation of runoff and thus, the transport of water and sediments..."

**Response to SC Line 65:** We have rephrased the sentence following these specific recommendations.

**SC Line 76:** You could make use of an abbreviation from line 14 here.

**Response to SC Line 76:** The term "reclaimed mining slope systems" is now abbreviated along the paper from this line: "*Mediterranean-dry reclaimed mining slope systems (hereafter RMSSs)*".

**SC Line 81:** to my understanding the routing of runoff is part of the structural connectivity while the processes which cause infiltration/excess of water to initialize, maintain or interrupt the flow of water is part of the functional hydrological connectivity.

**Response to SC Line 81:** Following the recommendations and for coherence with the applied terminology, we have reworded this sentence: *"the processes that initialize, maintain or interrupt the fluxes of water and sediments from the surface patch to the broader, hillslope scale"*.

**SC Line 84/88/91/95: Use abbreviation for the slope system.**

**Response to SC Line 84/88/91/95:** Following the recommendations, we have applied the abbreviation "RMSS" in these lines and throughout the paper.

SC Line 91: "transference of water..."

**Response to SC Line 91:** Following the recommendations, we have changed "transference of runoff" to "transference of water".

**SC Line (either) 101/129:** Add a short sentence like: "The field work was accomplished between October 2007 and November2008." After mentioning the dates of the survey no need for further repetition of the dates during the methods/results/discussion e.g. line 124/126....

**Response to SC Line (either) 101/129:** Following the recommendations, the fieldwork dates are now detailed in a single sentence located in the first lines of the Methods: *"The study site*

encompasses three experimental slopes, all north facing with a general gradient of about 20o (Figures 1b and 1d), which were surveyed intensively between October 2007 and December 2008".

**SC Line 103:** Sentence is incomplete and does not make sense.**

**Response to SC Line 103:** We have reworded the sentence: "Mean annual precipitation (MAP) is 450 mm, most of which occurs in spring and autumn. Potential evapotranspiration (PET; Hargreaves and Samani, 1985) is around 900 mm and the hydrological deficit (MAP-PET) is approx. 450 mm, concentrated in the summer months (López-Martín et al., 2007)".

**SC Line 105: Just: "Remarkable is..."**

**Response to SC Line 105:** We have rephrased the sentence following these specific recommendations.

**SC Line 115:** This sentence suggests that Slope 2 also has significant amounts of overland flow and erosion, which to my understanding is not the case.**

**Response to SC Line 115:** The berm that acts as a runoff contribution structure is much smaller in Slope 2 than in Slope 1, which limits its impact in the former as compared to the later. We have rephrased this sentence to avoid confusions: "These variations occurred due to the existence of a very steep (40o) and bare soil, runoff contributing berm integrated at the top of two of the experimental slopes (Slopes 1 and 2, with berm sizes of 50 and 20 m2, respectively; Figure 1b). This runoff contributing structure promoted soil erosion and conditioned the early dynamics of the experimental slopes, particularly in Slope 1, where the berm area is bigger and produces important amounts of overland flow".

**SC Line 121-125: This may also part of the results section.**

**Response to SC Line 121-125:** Although we understand the comment raised by the reviewer, we prefer to keep this information (gross runoff and sediment production of the slopes) in the description of the experimental site, as these numbers are already published in a previous paper (Merino-Martín et al. 2012a) and we do not apply any direct analysis of these gross variables (we just refer here to the published general information).

**SC Line 127:** Already mentioned that in the abstract and introduction. No need to have that long introduction here for the methods.

**Response to SC Line 127:** Following the recommendation, we have deleted the redundant information from the text.

**SC Line 130:** *"…Merino-Martin et al., 2012a), that included naturally delimited runoff/erosion plots distributed at the (i) hillslope and the (ii) surface-patch scale."*

Response to SC Line 130: We have rephrased the sentence following these specific recommendations.

**SC Line 136-139:** You mention Fig. 1d and 1f but not 1e. Usually the parts of the figures are described according to their alphabetic order. Either restructure the text or the figure.

**Response to SC Line 136-139:** Please, note that panel Fig. 1e is already cited in the text (line 145 of the original manuscript), although it is cited after Fig. 1f. We have re-ordered the citations in the text and the panels in Fig. 1 to describe them according to their alphabetic order.

**SC Line 140-143 & fig. 1f:** Categories for the species would increase direct readability of the figure. *E.g. Medicago sativa (Ms – A), Dactylis glomeratea (Dg - B), Santolina chamaecyparissus (Sch - B),...*

**Response to SC Line 140-143 & Fig. 1f:** We believe that increasing the complexity of the species abbreviations in the text with new characters has probably little interest in terms of increased readability of the paper. However, we agree with Referee # 1 that linking these species with the hillslopes in the figure can be useful. Therefore, we have modified panel (f) of figure 1 to link the dominant species with the hillslopes using the following labels/codes:

Ms - *M. sativa* (Slopes S1, S2)

Lp - *L. perenne* (Slopes S1, S2)

Sch - *S. chamaecyp.* (Slopes S1, S2, S3)

Dg - *D. glomerata* (Slope S2)

Br - *B. retususm* (Slope S2)

Tv - *T. vulgaris* (Slopes S2, S3)

Gs - *G. scorpius* (Slopes S2, S3)

**SC Line 150-151:** Sedimentological methods, may adjust header of the chapter to field measurements.

**Response to SC Line 150-151:** We have adjusted the header of the (sub)section to: "2.2 Field acquisition methods of hydro-sedimentary and precipitation data".

**SC Line 153-169:** *Climatological, soil hydrological and statistical analytical methods mixed. I suggest* to split the statistical part from the pure data acquisition part. A table showing an overview of the climatological statistics would be beneficial also for the introduction of the predictors for the GLM later on.

**Response to SC Line 153-169:** Following the recommendations, we have split the statistical and data acquisition information and generated a new section in the Methods for the description of the statistical methods (section "2.5 Data analysis and statistics"). In addition, we have added the following new table (new Table 1) that shows an overview of the precipitation condition variables applied in the study:

|                 | Description                    | Units              |
|-----------------|--------------------------------|--------------------|
| Dp              | Storm depth                    | mm                 |
| Rd              | Rainfall duration              | h                  |
| I 15 | 15-min max rainfall intensity  | mm h⁻¹             |
| I 30 | 30-min max rainfall intensity  | mm h -1 |
| Im              | Mean rainfall intensity        | mm h⁻¹             |
| API             | Antecedent precipitation index | mm                 |

**SC Line 171:** *"Previous research carried out..."* (*References missing!*).

**Response to SC Line 171:** Although the reference was already included in the original version of the paper (line 175: Moreno-de-las-Heras et al., 2009), we agree with the referee that the first line of the information is a better place to the citation in the text: "*Previous research carried out in the Utrillas field site applying small-scale (0.25 m2) rainfall simulations (Moreno-de-las-Heras et al., 2009) (...)".*

**SC Line 172:** Why using a range here when a non-dynamic threshold of 50% is applied?

**Response to SC Line 172:** We have modified the text, in line with the applied non-dynamic threshold of vegetation cover: *"surface patches with vegetation cover under 50% can generate important amounts of runoff/sediments"*.

SC Line 179: (0.5m resolution)

Response to SC Line 179: We have added the suggested change in the text.

**SC Line 184:** *"To this end," is a fill word and can be deleted.*

**Response to SC Line 184:** Following the recommendations, we have removed the expression "to this end".

**SC Line 199:** Maybe better: "...until a sink (i.e. >50% vegetation cover) or the outlet of the system is reached." And "outlet of the system is reached" is unclear which system patch or hillslope?

**Response to SC Line 199:** Following the recommendations, we have reworded this sentence: "*until a sink (i.e., >50% vegetation cover) or the outlet of the hillslope is reached*".

**SC Line 199(b):** In general, introducing a figure to illustrate the different steps of the calculation and also the use of mathematical symbols and equations to clarify the calculated ratio in line 201 could help to increase understanding for the reader.

**Response to SC Line 199(b):** We agree with Referee # 1 that illustrating the different steps of the calculations in a figure can strongly facilitate understanding of the calculated index of structural connectivity. We have added the following schematic figure (new Fig. 2a) illustrated with a virtual example that is linked to the text using fully explicit mathematical symbols and equations: